# Evaluating privacy risks of parameter-efficient fine-tuning

## Abstract

Parameter-efficient fine-tuning (PEFT) is a new paradigm for fine-tuning language models at scale. Unlike standard fine-tuning, PEFT adjusts only a small number of parameters, making it more computationally accessible and enabling practitioners to develop personalized services by fine-tuning models on user data. Because the models are trained on user data, this emerging paradigm may attract adversaries who want to extract sensitive information from fine-tuning data. However, to date, their privacy implications have not been well-understood yet in the literature.

In this paper, we study the impact of this new fine-tuning paradigm on privacy. We use an off-the-shelf data extraction attack as a vehicle to evaluate the privacy risk on two pre-trained language models fine-tuned on 2 datasets, repeated 5 times with different random seeds, resulting in a total of 100 variations. Our main findings are: (1) for practitioners employing PEFT to construct personalized models, the fine-tuned models have lower privacy risks while maintaining reasonable utility; (2) for developers designing new PEFT algorithms, while safer than standard fine-tuning, certain design choices in the algorithms increases memorization in an unexpected way; and (3) for researchers auditing the privacy of fine-tuned models, employing weak differential privacy is sufficient to mitigate existing data extraction risks without significantly compromising model utility. We hope our work encourages the safe adoption and development of PEFT algorithms in practice, as well as future work on advancing stronger privacy auditing mechanisms.

## 1 Introduction

"Pre-training and fine-tuning" is a common paradigm in developing AI services built on commercial-scale language models. Model providers like Google[1], Meta[2], or OpenAI[3] handle the pre-training stage, while service providers fine-tune the ready-made models on their own datasets. Because those models have a large number of parameters, the fine-tuning process requires extensive computational resources. As a potential solution, there has been active research on reducing these computational demands, such as parameter-efficient fine-tuning (PEFT) (Han et al., 2023).

Against this common paradigm, recent work has demonstrated *data extraction attacks* (Carlini et al., 2023). To breach the confidentiality of AI services, an adversary exploits the model's query interfaces to reconstruct training data from the fine-tuned models. Given that the data used for fine-tuning likely includes private records of service users, this poses a significant privacy risk, with models potentially leaking personally identifiable information (PII), such as patient names or email addresses.

In this work, we study the risk of data extraction attacks given rise to by the emerging paradigm: PEFT. Most work on data extraction targets pre-trained models as-is (Carlini et al., 2019; 2021; 2023; Nasr et al., 2023) or focuses on scenarios where the entire parameters are fine-tuned (Ponomareva et al., 2022; Jayaraman et al., 2024). However, it remains unknown how vulnerable these fine-tuned models, especially those constructed using PEFT algorithms, are to data extraction attacks. It is also unclear which design choices in PEFT algorithms make them more (or less) vulnera-

---

[1]https://cloud.google.com/vertex-ai/generative-ai/docs/models/tune-models

[2]https://www.llama.com/docs/how-to-guides/fine-tuning

[3]https://platform.openai.com/docs/guides/fine-tuning

ble to data extraction attacks. Moreover, it is essential to understand how the formal defense against privacy attacks—differential privacy—mitigate this risk while maintaining model utility.

**Contributions.** We *first* address these questions by comprehensively evaluating the privacy risks of language models fine-tuned with various PEFT algorithms. We use an off-the-shelf data extraction attack, developed by (Carlini et al., 2019), as a vehicle to assess this privacy threat. We fine-tune two commercial-scale language models using five different fine-tuning algorithms on two datasets repeated five times with different random seeds, to achieve 80 variations of PEFT-trained models, and 20 with full-model finetuning.

We demonstrate that models constructed using PEFT algorithms achieve 2–14× times less exposure, while standard fine-tuning leads to the successful extraction of secrets from the resulting models. We also observe variations in memorization across models fine-tuned with different PEFT algorithms.

*Second*, we characterize key factors that influence the memorization of secrets across different fine-tuning algorithms. We show that secrets containing substrings likely to appear in the pre-training corpus are less likely to be memorized by fine-tuned models. In contrast to the prior work's findings, we observe that the increase in the number of tunable parameters does *not* necessarily mean more memorization in models. Moreover, we find that certain design choices in PEFT algorithms can lead to different memorization patterns. In prefix-tuning, for example, secrets located at the beginning of a training record are more easily memorized than those placed at the end.

*Third*, we investigate the interaction between a privacy defense with the formal guarantee (differential privacy, $\epsilon$) and model utility across five fine-tuning algorithms. We demonstrate that, even with a large $\epsilon$, data extraction can be completely rendered ineffective across all PEFT algorithms, while preserving model utility. One can also reduce $\epsilon$ to 2.0–5.0, depending on the PEFT algorithm used, without significant performance loss. We find that PEFT algorithms that fine-tune fewer parameters are better at preserving model utility under strong privacy guarantees, $\epsilon \in [0.2, 2.0]$. We also show that lower ranks are preferable for keeping model utility under small $\epsilon$ values.

We hope our work will serve as a Hitchhiker's Guide to fine-tuning language models with privacy.

## 2 BACKGROUND AND RELATED WORK

**Parameter-efficient fine-tuning (PEFT)** enables to fine-tune large-scale models in a computationally accessible way while maintaining performance comparable to standard fine-tuning. Instead of adjusting the entire model parameters, PEFT reduces the number of tunable parameters through various methods Han et al. (2024). A common approach is to use additive methods: we alter the model architectures by injecting small learnable modules (or parameters). Representative methods include (1) *adapters* (Houlsby et al., 2019) where small learnable modules are added to transformer blocks; (2) *prefix-tuning* (Li & Liang, 2021), which introduces learnable vectors added to keys and values across all transformer layers; and (3) *prompt-tuning* (Lester et al., 2021) that applies learnable vectors only at the initial token embedding layer to enhance training and inference efficiency. An alternative yet emerging approach is Low-Rank Adaptation (LoRA) Hu et al. (2022), which constructs a low-rank parameterization of transformer layers to reduce the number of tunable parameters. Our work studies memorization of models fine-tuned though these PEFT algorithms. Concurrently, Anonymous (2024) studies tight auditing of memorization in standard fine-tuning. But our focus is more on the impact of memorization under these emerging fine-tuning techniques.

**Privacy risks in language model ecosystem.** Data extraction attacks present a major risk to the language model ecosystem: an adversary aims to extract private information from the training data used to train (or *fine-tune*) language models. Because language models are deployed in a black-box manner, most prior attacks have demonstrated their feasibility by exploiting query-based interactions. Initial work on data extraction focuses on extracting private information, unintentionally *memorized* during pre-training (Carlini et al., 2019; 2021; Nasr et al., 2023; Carlini et al., 2023; Bai et al., 2024), but as fine-tuning becomes more common, recent work explores the extraction of sensitive data from fine-tuning data (Lukas et al., 2023; Liu et al., 2024). Our work falls into the latter category, as we study data extraction against fine-tuned models, which is under-explored in the prior work.

How precisely an attacker queries the target model varies depending on their knowledge. The weakest attacker has only query access to the target model and no knowledge of the training data. This

attacker will choose prompts that are likely to trigger the generation of memorized data, which may take forms, such as random Internet strings Carlini et al. (2021); Nasr et al. (2023) or special characters (Bai et al., 2024). These attacks are untargeted, aiming to reconstruct any training examples verbatim. On the other hand, a strong adversary has (partial) access to the training data and knows the context associated with private information. The adversary can prompt the target model using these prefixes to reconstruct the remaining specific tokens in the training records to which the prefix belongs (Carlini et al., 2023; Lukas et al., 2023). Because our work uses data extraction attacks as a privacy auditing mechanism, we perform a membership-inference style attack, where the adversary knows the context associated with a secret and has a list of secret candidates to compare.

**Differential privacy (DP)** (Dwork et al., 2006) is originally developed to reduce the difference in outcome from querying two databases which differ by a single record. Abadi et al. (2016) developed a training algorithm, differentially-private stochastic gradient descent (DP-SGD), that employs DP to guarantee protection of a model against the worst-case private information leakage. DP-SGD formally quantifies the leakage with the parameter $\epsilon$. We set $\epsilon$ to a desired value before training, and once the total leakage exceeds the pre-defined $\epsilon$ during training, we stop training and save the model with its parameters. To date, DP-SGD is the standard practice for training (or fine-tuning) private models (Ponomareva et al., 2022; Li et al., 2022; Yu et al., 2022). However, the privacy guarantee comes at the cost of performance: a stronger guarantee often results in significant performance degradation. Thus, it is important to understand the privacy-utility trade-off (Jayaraman & Evans, 2019) and how to train private models with performance comparable to non-private models (Ponomareva et al., 2023). Our work also studies the privacy-utility trade-off in fine-tuned models.

A separate line of work studies defensive mechanisms in the context of language models to mitigate *empirical* privacy risks. Deduplication reduces the number of secret occurrences in the training data to mitigate data extraction attacks (Kandpal et al., 2022; Lee et al., 2022). Adversarial training (Goodfellow et al., 2015), a standard countermeasure against adversarial examples, is used with a privacy regularizer to jointly optimize for both privacy and utility (Mireshghallah et al., 2021). While these defenses effectively reduce the success rate of existing privacy attacks (Rigaki & Garcia, 2023), we exclude them from our investigation as they do not provide formal guarantees.

## 3 METHODOLOGY

### 3.1 DEFINITION OF MEMORIZATION

We adopt the definition of memorization from Carlini et al. (2023), with adaptations in blue.

**Definition 3.1. (Memorization)** A secret $s$ is memorized by a model $f$ with $k$ tokens of context if there exists a (length-$k$) string $p$, such that the insertion of $s$ into $p$, denoted as $p \oplus s$ is present in the training data for $f$, and $f$ achieves the lowest perplexity, when prompted with $p \oplus c$ where $c$ is $s$, across all possible secret candidates $c$ in $C$.

This definition differs from prior work (Carlini et al., 2021; 2023; Nasr et al., 2023; Bai et al., 2024). Instead of prompting the model with a context $p$ and then generating next $N$ tokens using a given decoding method, we compute the *perplexity* directly on a list of prompts $p \oplus c$, each differing only by the secret candidate $c$. The difference lies in the purpose of employing data extraction. Prior work focuses on demonstrating the feasibility of data extraction against language models in use, but we leverage the same attack for auditing privacy risks. Hence, rather than prompting the model with $p$ and hoping greedy decoding extracts the correct final token, we assume a strong adversary by limiting their search space to a set of candidate secret tokens $C$, expecting the true secret $s$ to yield the lowest perplexity compared to the others. One can view this definition as an edge-case of Carlini et al. (2023), where the attacker has all but the final token of a training record.

### 3.2 QUANTIFYING MEMORIZATION

**Threat model.** We consider an emerging scenario where a victim develops natural language processing services by fine-tuning a commercial-scale pre-trained language model on their data, which may contain private information of users. Because these models have more than billions of parameters, we assume that the victim employs PEFT methods to reduce the computational demands for fine-tuning. We assume a data extraction adversary (Carlini et al., 2021; 2023; Nasr et al., 2023; Bai

et al., 2024; Lukas et al., 2023), also an emerging concern to language model ecosystem, who aims to extract private information from a target model. In our scenario, we assume an oracle adversary with *black-box* access, exploiting the model's prompting interface.

**Exposure as a metric for quantifying memorization.** Our definition above is *strict*: memorization is only confirmed when the prompt containing the secret achieves the lowest perplexity. However, in our initial investigation, we find the need to *relax* this definition slightly. While the strict definition is useful for determining the success of an attack, it does not provide a measure of the degree to which a secret is memorized by a model. In consequence, in most cases where the perplexity is not the lowest (even when the value is a close second or runner-up), it is considered as not-memorized.

**Definition 3.2. (Exposure)** Given a secret $s$ and a model $f$, the exposure of $s$ is defined as:

$$\mathbf{exposure}_f(s) = \log_2 |C| - \log_2 \mathbf{rank}_f(s)$$

We follow the definition of Carlini et al. (2019). The cardinarlity of the candidate space $C$, is set to approximately 400. The **rank** of a secret $s$ is defined as its index in the list of all possible candidates in $C$, ordered by the model perplexity. In our case, the "candidate space $C$" refers to the number of possible candidates a secret $s$ could be, instead of every possible character combinations with the same length as $s$. We make this decision for computationally practical threat modeling. In the medical record dataset (MIMIC) we use, a 10-character secret, such as a patient's name in English, has $27^{10}$ combinations. But we reduce the space to $400$, by selecting only common English names.

### 3.3 PREPARING THE EVALUATION DATA

We prepare two different types of datasets for our evaluation. The first dataset represents the most challenging scenario for our data extraction adversary: *a single insertion* of a secret $s$. In this case, we randomly select a record $p$ from the training data and concatenate the secret, forming $[p||s]$. This construction follows the same methodology as in Carlini et al. (2023). We take a dataset and repeat this process five times with different random seeds to construct five distinct fine-tuning datasets.

While commonly used in the literature, the previous construction may not capture the variations in the secret's location within a context. For instance, when the secret is a patient's name, the training record could be "John Doe is diagnosed with granulosa cell tumor" rather than "Granulosa cell tumor is the disease for John Doe." To study the impact of a secret's location on memorization, we select 50-token-length training records from a dataset, insert the same secret at 5 different positions, and save each version as a separate fine-tuning dataset. We also examine how duplication affects memorization by increasing the number of duplications from 1 to 500 for each fine-tuning dataset.

## 4 EMPIRICAL EVALUATION

### 4.1 EXPERIMENTAL SETUP

**Datasets.** We fine-tune models using two datasets: MIMIC-III (Johnson et al., 2016) and the Enron corpus[4]. The MIMIC-III dataset contains 112,000 de-identified electronic health records, including vital signs, lab results, and patient status reports. Due to the size complexity, we sample a subset of the entire data, focusing on 13,431 records of patient bedside checkups. The Enron email corpus, widely used in data extraction research (Carlini et al., 2019; Lukas et al., 2023), contains over 600,000 emails exchanged between Enron Corporation employees, collected by the Federal Energy Regulatory Commission during its investigation. We use it to ensure comparable and generalizable findings. We extract a subset of 13,399 records to match the size of our MIMIC dataset.

**Secrets.** We insert a synthetic patient name "mary smith," once into the MIMIC-III dataset, and a faux email address, "Leo.Moreno@gmail.com," into the Enron corpus. This testing strategy is similar to the prior work (Jayaraman et al., 2024; Liu et al., 2024), where artificial secrets are inserted into training datasets. In order to compute exposure, we also prepare 400 additional secret candidates using other common names and emails, such as "james henderson" or "Maria.Hernandez@yahoo.com." Please refer to Appendix B.13 for example records we insert.

---

[4]https://www.cs.cmu.edu/ enron/

**Models.** We use autoregressive models, GPT-2 and GPT-2 XL (Radford et al., 2019), in our experiments, as these models are widely employed in data extraction research and are predecessors of commercial-scale language models like GPT-4 (Achiam et al., 2023). GPT-2 is a decoder-only transformer model with 124M parameters, while GPT-2 XL is a production-scale version of GPT-2, with $4\times$ the number of layers, and $\sim 2\times$ the parameters per layer, resulting in a total of 1.5 billion parameters. Please refer to the Appendix A for details on our fine-tuning hyperparameter selections.

**Metrics.** As described in Sec 3.2, we compute exposure to quantify memorization of a secret by fine-tuned models. To measure the performance of these models, we compute perplexity, the exponential of the model loss over a given sequence, on the evaluation data.

## 4.2 MEMORIZATION OF FINE-TUNED MODELS

We first compare the memorization of a secret across models fine-tuned using standard fine-tuning and four PEFT methods: fine-tuning with Adapters, Prefix-tuning, Prompt-tuning, and LoRA.

| | | | | PEFT Method | | | |
|---|---|---|---|---|---|---|---|
| **Dataset** | **Models** | **Metric** | **Baseline** | **Adapter** | **Prefix-tuning** | **Prompt-tuning** | **LoRA** |
| MIMIC-III | GPT-2 | Exp. | 8.64±0.00 | 3.71±0.97 | 3.72±1.46 | 2.70±0.41 | **1.88±1.25** |
| | | PPL. | **1.15±0.00** | 1.30±0.01 | 1.24±0.00 | 1.23±0.00 | 1.17±0.00 |
| | GPT-2 XL | Exp. | 8.64±0.00 | 4.46±0.28 | 4.48±1.18 | **1.51±0.56** | 5.29±1.01 |
| | | PPL. | 1.15±0.00 | 1.30±0.00 | 1.27±0.01 | 1.20±0.00 | **1.13±0.00** |
| | Pythia-2.8B | Exp. | 8.64±0.00 | 2.69±1.54 | 1.56±0.04 | **0.89±0.16** | 2.83±2.21 |
| | | PPL. | 1.16±0.00 | 1.12±0.00 | 1.27±0.01 | 1.16±0.00 | **1.12±0.00** |
| Enron | GPT-2 | Exp. | 8.04±1.20 | 1.47±0.68 | 0.55±0.34 | **0.45±0.31** | 1.28±0.76 |
| | | PPL. | 1.08±0.00 | 1.18±0.01 | 1.11±0.01 | 1.12±0.01 | **1.06±0.00** |
| | GPT-2 XL | Exp. | 7.63±1.58 | 1.35±0.57 | 0.86±0.53 | 0.73±0.50 | **0.50±0.34** |
| | | PPL. | **1.06±0.00** | 1.21±0.00 | 1.16±0.01 | 1.13±0.01 | 1.07±0.00 |
| | Pythia-2.8B | Exp. | 8.64±0.00 | **1.05±1.03** | 1.95±0.03 | 1.17±0.87 | 1.52±0.51 |
| | | PPL. | 1.07±0.00 | 1.05±0.00 | 1.18±0.00 | 1.07±0.00 | **1.05±0.00** |

Table 1: **Comparison of data extraction success across language models.** We compute the exposure (Exp.) and the evaluation perplexity (PPL.) of language models fine-tuned using six different algorithms. Each cell reports the average over five runs along with the standard deviation. In each case, the secret is inserted once into the fine-tune dataset. We bold the lowest evaluation perplexity, as well as lowest exposure for each model-dataset pair.

**Results.** Table 1 summarizes our results. We find that the models fine-tuned through PEFT algorithms are less vulnerable to data extraction. Standard fine-tuning (Baseline) results in the exposure values close to maximum ($\sim 8.64 = \log_2 401$), but when we employ PEFT algorithms, the exposures are reduced by $2$–$14\times$ times ($0.50$–$4.46$). We also compare the perplexity of fine-tuned models to verify that the reduction is not from the performance loss. We observe a slight increase in perplexity ($0.01$—$0.15$), but the increase is too small to result in a significant decrease in the exposure. We also show in Appendix B.11 that the reduction in exposure is not due to performance degradation. Even with the comparable perplexity (see LoRA columns), we find the exposure is reduced by $14\times$ times.

Note that we fix the number of training epochs we use for fine-tuning, e.g., fine-tuning of GPT-2 on Enron uses 10 epochs, to reflect the real-world training practice. If we increase the number of epochs to 50 and beyond, while there is no performance benefit, the exposure values computed on the fine-tuned models increase. But we do not evaluate such cases, as there is no reason a victim will use $5\times$ more training epochs with computationally-efficient fine-tuning algorithms.

Across the different PEFT methods, we find that prompt-tuning and LoRA consistently demonstrate the lowest exposure values. In prompt-tuning, we attribute the low exposure to the type of parameters are trained. While other PEFT mechanisms tunes the parameters across all the transformer layers, including the attention and the fully-connected layers, prompt-tuning only fine-tunes the subset of a model's embedding layers. Due to this design choice, prompt-tuning may make it more difficult for the model to associate a secret with various contexts in the training data. In LoRA, the reduced rank in the latent representation space acts as an information bottleneck, making it difficult for the

model to memorize outliers, such as the secret, which the model first encounters during fine-tuning (as we ensure the secret is not present in the pre-training corpus; see Appendix B.8). Please refer to Appendix B.12 for a detailed investigation of our hypothesis.

## 4.3 FACTORS INFLUENCING MEMORIZATION IN FINE-TUNING

We now shift our focus to the factors influencing memorization during fine-tuning. This includes our experimental design, such as the datasets and secrets we use, or the PEFT hyper-parameters.

**Impact of the secret types.** In our experiments (Table 1), the reduction in the exposures in Enron are greater than that observed in MIMIC-III. We attribute this difference to the secrets we choose. In MIMIC-III, we use a patient name with medical records; both the models pre-trained on the curated Internet sources are not likely to encounter medical records. The memorization of the patient name may be easier than that of the secret we use in Enron—a synthetic Gmail address that the pre-training data corpus is likely to contain. During fine-tuning, it could be difficult for the model to distinguish our secret email address from the other Gmail addresses learned from the pre-training step. We run additional experiments with rare names and email addresses as secrets in the MIMIC dataset and make consistent observations. The details can be found in Appendix B.10.

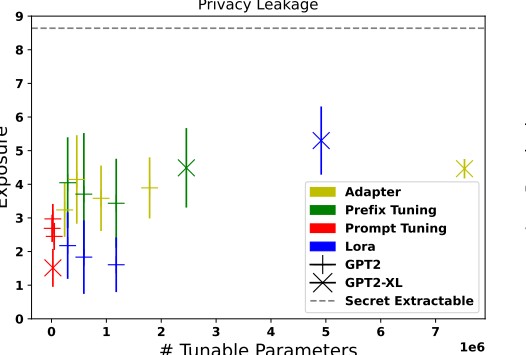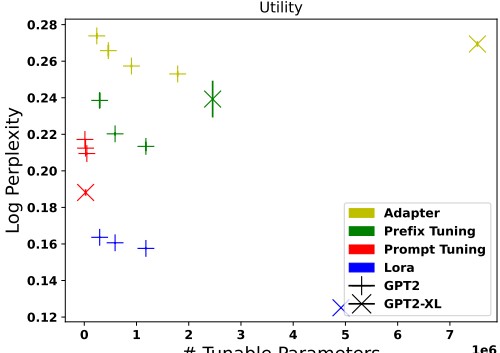

Figure 1: **Impact of tunable parameter count on memorization.** On the left, we compare the exposure of fine-tuned models with varying number of tunable parameters. We also show the evaluation perplexity of these models on the right. We run this evaluation on MIMIC-III.

**Impact of tunable parameter counts.** Prior work has demonstrated that increasing the number of tunable parameters leads to greater memorization (Carlini et al., 2023). This holds true at scale: standard fine-tuning of GPT-2 and GPT-2 XL models results in perfect memorization of a secret—even when the secret appears only once in the fine-tuning data. However, it remains under-explored whether this observation holds in the context of PEFT. To evaluate this hypothesis, we compare the exposure in fine-tuned models based on the number of parameters tuned by each PEFT algorithm.

Figure 1 summarizes our results in MIMIC-III. We have consistent findings from our Enron experiments (refer to Appendix B.1 for our Enron results). We compare the difference between fine-tuned GPT-2 and GPT-2 XL models. Because GPT-2 XL have more parameters than GPT-2, applying PEFT algorithms to GPT-2 XL result in tuning more parameters during fine-tuning. Prior work's findings are not consistent with our observations across different PEFT algorithms. In both Adapter and LoRA, fine-tuned GPT-2 XL models exhibit higher exposure values, as expected. However, we do *not* observe any significant differences in prefix tuning. Surprisingly, we find GPT-2 XL models fine-tuned with prompt tuning exhibit exposure values lower than GPT-2 models.

One possiblity is that a smaller number of tunable parameters could lead to performance degradation in fine-tuned models for the task at hand. To analyze further, we compare the evaluation perplexity across various fine-tuned models. In LoRA, our result aligns with existing knowledge: an increase in the number of tunable parameters reduces evaluation perplexity, which unintentionally leads to the increase in memorization of secrets. However, in other three PEFT techniques, we observe the decrease in exposure as the model become accurate on a desired task.

## 4.4 DOES THE POSITION OF A SECRET WITHIN A SENTENCE MATTER?

Most prior work follows the definition of memorization from (Carlini et al., 2019; 2023), where a secret $s$ is concatenated at the end of a context $p$. Now we challenge this practice and analyze further how the position of a secret within a context impacts memorization. Our hypothesis is that PEFT methods, which only tune parameters corresponding to specific token positions in the input, may be better at memorizing secrets in those locations than secrets placed at the end. Here we focus on our findings in MIMIC-III. Please refer to Appendix B.3, B.5, and B.6 for our full results.

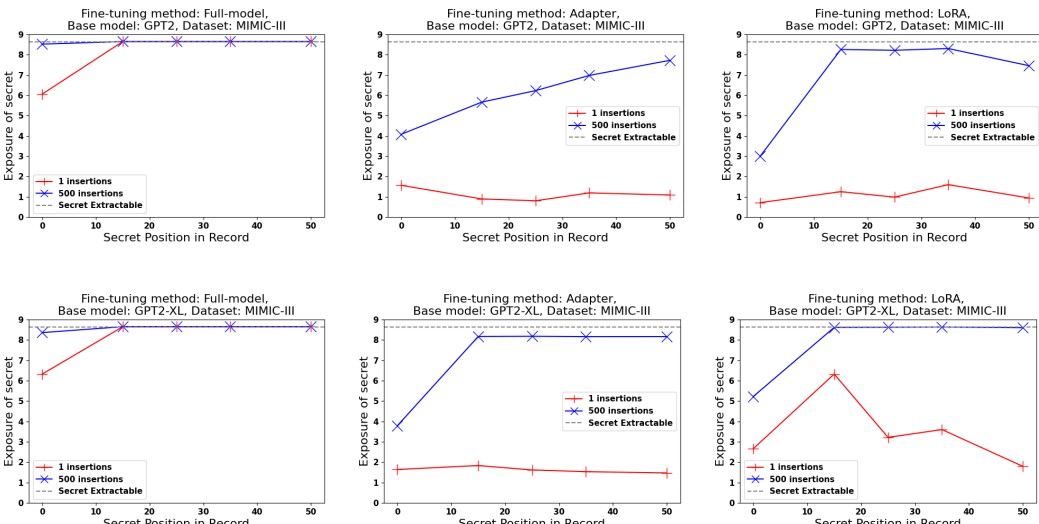

Figure 2: **Illustrating the impact of secret position on memorization.** The figures show the impact of a secret's location in a context on exposure. The top row shows the results from GPT-2 models, while the bottom row presents results from GPT-2 XL. From the left, each column corresponds to standard fine-tuning, fine-tuning with adapters, and LoRA. We show the results on MIMIC-III.

**On standard fine-tuning, fine-tuning with adapters, and LoRA.** Figure 2 illustrates our findings in MIMIC-III. We first observe that when a secret is inserted only once in the fine-tuning data, there is no discernible impact on the secret's exposure across the three methods. However, when the number of insertions is increased to 500, we observe that secrets are more easily memorized if they appear in later positions within the target context, particularly when fine-tuning with adapters and LoRA are employed. Our observation align with prior work (Carlini et al., 2023): due to the autoregressive nature of modern language models, tokens in later positions within a sequence are more likely to be memorized.

**On prompt-tuning.** We observe in prompt-tuning consistently low exposure across the dataset and secret positions (less than ∼2.0). We also find no significant increase in exposure when the number of secret insertions is increased from 1 to 500. While prompt-tuning fine-tunes a few parameters at the earlier positions in prompts, it does not imply that the method can effectively memorize secrets in those positions. Prompt-tuning adds virtual tokens (or virtual prefixes) to each training record and tunes only the corresponding embedding layers. Thus, even if we place secrets in earlier positions of our training records, virtual tokens introduced by prompt-tuning will always be preceding. Please refer to Appendix B.7 for our full results on prompt-tuning.

**On prefix-tuning.** An interesting observation from our prefix-tuning experiments is that secrets located at the beginning of training records are more likely to be memorized. Figure 3 shows this observation. The left figure shows results from models trained without differential privacy (DP), while the right figure presents results with DP at $\epsilon$=10.0. Note that due to the space constraints, we include the DP results in Figure 3. When a single secret is inserted into the fine-tuning data, exposure slightly decreases as the secret's position within the target record moves to later locations. This trend becomes more distinct when 500 secrets are inserted into the fine-tuning data. In the left figure, the exposure at position 1 is ∼3, while it decreases to ∼2 at position 50. Note that exposure is measured on a log-scale; a decrease of 1 in exposure equals to a $2\times$ reduction in privacy risk.

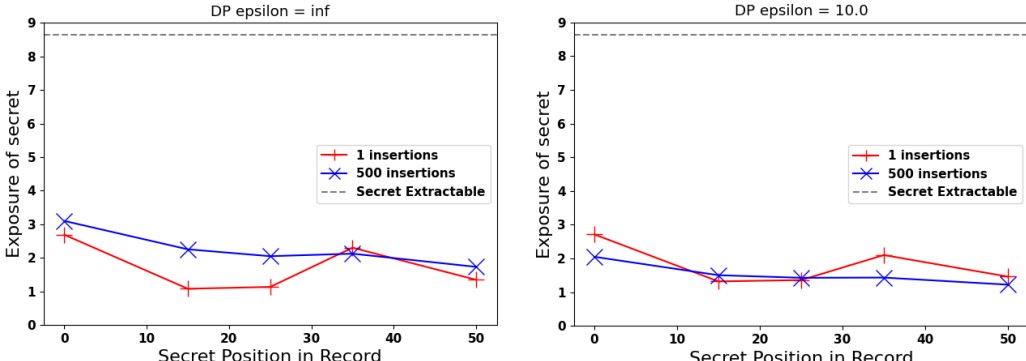

Figure 3: **Prefix tuning memorizes secret closer to beginning of record better.** In these figures, we show the effect of secret position in record vs. exposure when using the prefix-tuning, without DP ($\epsilon = \inf$; left) and $\epsilon = 10.0$ (right). We run this evaluation on GPT-2 in MIMIC-III.

## 4.5 MEMORIZATION OF MODELS FINE-TUNED WITH PRIVACY

We further test how the standard practice in training models with a privacy guarantee, differentially-private (DP) model training (Abadi et al., 2016), interacts with the four PEFT methods. We fine-tune both GPT-2 models using standard fine-tuning and four PEFT methods with varying epsilons in $\{0.01, 0.05, 0.075, 0.1, 0.5, 1.0, 2.0, 4.0, 8.0, 10.0\}$. For GPT-2, we run fine-tuning five times with different random seeds, but due to the resource limits, we fine-tune GPT-2 XL only once on $\epsilon$ of 0.1. We fine-tune for the same number of epochs as in the non-DP setting, ensuring a low, comparable evaluation perplexity reached at a loose privacy guarantee ($\epsilon$ of 10.0). We use the FastDP library (Bu et al., 2024), compatible with all four PEFT algorithms we employ.

| Method | Metric | Privacy Budget ($\epsilon$) | | | | | | | |
|---|---|---|---|---|---|---|---|---|---|
| | | $\infty$ | 10.0 | 8.0 | 4.0 | 2.0 | 1.0 | 0.5 | 0.1 |
| **Baseline** | Exp. | 8.64±0.00 | 2.20±1.78 | 2.21±1.78 | 2.34±1.64 | 2.50±1.24 | 2.47±1.00 | 2.41±0.95 | 1.75±0.66 |
| | PPL. | 1.15±0.00 | 1.12±0.00 | 0.12±0.00 | 1.12±0.00 | 1.13±0.00 | 1.13±0.00 | 1.13±0.00 | 1.15±0.00 |
| **Adapter** | Exp. | 3.71±0.00 | 2.94±0.92 | 3.28±1.57 | 3.00±2.07 | 3.36±1.60 | 2.94±1.98 | 2.65±1.75 | 2.10±1.32 |
| | PPL. | 1.30±0.01 | 1.42±0.01 | 1.43±0.00 | 1.46±0.02 | 1.59±0.11 | 1.63±0.11 | 1.78±0.25 | 5.43±2.79 |
| **Prefix-tuning** | Exp. | 3.72±1.46 | 3.22±1.03 | 3.16±1.02 | 3.24±1.22 | 3.15±1.24 | 3.18±1.15 | 3.27±0.10 | 2.83±0.91 |
| | PPL. | 1.24±0.00 | 10.36±12.36 | 13.74±17.01 | 24.44±24.80 | 43.35±32.94 | 73.42±44.06 | 127.94±61.56 | 815.65±800.74 |
| **Prompt-tuning** | Exp. | 2.70±0.41 | 1.99±0.51 | 2.00±0.53 | 2.02±0.54 | 2.00±0.57 | 2.01±0.58 | 1.98±0.60 | 1.96±0.60 |
| | PPL. | 1.23±0.00 | 1.92±0.03 | 2.45±0.07 | 11.43±1.06 | 70.75±2.24 | 202.32±2.18 | 438.74±3.92 | 1448.78±10.66 |
| **LoRA** | Exp. | 1.88±1.25 | 2.68±0.85 | 2.70±0.87 | 2.74±0.96 | 2.72±0.97 | 2.63±0.95 | 2.57±0.91 | 2.16±0.30 |
| | PPL. | 1.17±0.00 | 1.20±0.00 | 1.20±0.00 | 1.21±0.00 | 1.21±0.00 | 1.21±0.00 | 1.22±0.00 | 1.28±0.00 |

Table 2: **Comparison of DP epsilon against exposure and perplexity.** We compute the exposure (Exp.) and the evaluation perplexity (PPL.) of language models fine-tuned using five different finetuning methods for eight different DP epsilons (including without any privacy - $\infty$). Each cell reports the average over five runs along with the standard deviation.

**Memorization vs. perplexity.** We begin with comparing the impact of different privacy guarantees on empirical privacy risks (measured as exposure) and model performance (measured as evaluation perplexity). In evaluation, we set the adapter rank to 32, the number of prompt and prefix tokens both to 64, and the LoRA rank to 16. We find that $\epsilon$ values below 10.0 render data extraction attacks completely ineffective. At $\epsilon = 10.0$, we observe exposure values between 2 and 3, a $4\times$ reduction in exposure compared to standard fine-tuning without DP, indicating that the secrets rank between the 50th and 100th positions in the list of candidates, ordered by evaluation perplexity. Most PEFT methods do not result in significant performance degradation, except for prefix-tuning, which achieves an evaluation perplexity of approximately 5 at $\epsilon = 10.0$. Setting $\epsilon$ below 10.0 completely breaks the models fine-tuned with prompt tuning and prefix tuning, with their perplexity exceeding 300. LoRA models achieve the best exposure-perplexity trade-off. Our results are consistent with GPT-2 XL models. Please refer to Appendix B.6 for our full results.

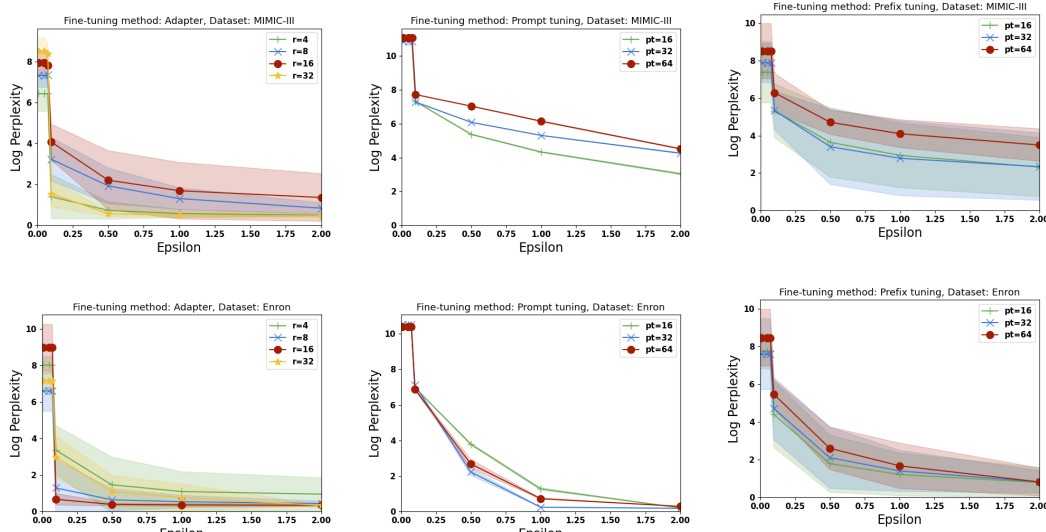

Figure 4: **Impact of privacy guarantee $\epsilon$ on model perplexity.** We illustrate the trade-off between $\epsilon$ and evaluation perplexity, measured on our fine-tuned GPT-2 models. (from the left) We show the results from fine-tuning with adapter, prompt-tuning, and prefix-tuning with different configurations. MIMIC-III datasets are located on top and Enron datasets below.

**Trade-off between privacy and utility.** Next we analyze the tradeoff between privacy, guaranteed formally by $\epsilon$, and utility (measured by evaluation perplexity). Figure 4 summarizes our results from GPT-2 models fine-tuned on MIMIC-III and Enron datasets. We use $\epsilon$ in [0.001, 2.0] and explore the impact of different PEFT hyperparameters: with the adapter ranks in $\{4, 8, 16, 32\}$, the number of prompt and prefix tokens in $\{16, 32, 64\}$, and the LoRA ranks in $\{8, 16, 32\}$. We focus on a reduced epsilon range, $\epsilon \in [0.1, 2.0]$, as perplexity increases by orders of magnitude within this range. If we use $\epsilon < 0.1$, the fine-tuned models perform no better than random.

Overall, we observe a greater increase in perplexity as we increase the configuration values across PEFT algorithms. This occurs because the configurations are proportional to the number of tunable parameters: an increase in tunable parameters requires adding more noise to achieve the same target $\epsilon$ value as when fine-tuning models with fewer tunable parameters. For adapters, we observe an increase in perplexity from ~1.1 to ~8.0 at $\epsilon = 0.1$, whereas the increase reaches up to ~800–2400 for the case of prompt-tuning and prefix-tuning. The notable exception is LoRA, not shown in the above figure, which impressively maintains low-perplexity even at $\epsilon = 0.1$. Our results indicate that once a sufficiently low-perplexity is achieved with a PEFT method's configuration, increasing the number of tunable parameters can lead to a worse trade-off between privacy and utility. We further investigate the privacy-utility trade-off with the larger GPT2-XL base model in Appendix B.6

However, we do not observe any consistent relationship between PEFT hyper-parameters and the perplexity under DP. The first observation we had is that the trend differs from the datasets we use. In the MIMIC-III dataset, larger hyperparameters generally result in higher perplexity (except for Adapter with $r = 32$). In contrast, we observe a reduction in perplexity for the Enron dataset under the same conditions. We hypothesize that there are optimal PEFT hyper-parameters required to achieve reasonable performance (e.g., in MIMIC-III, the rank ~4–8 and the prefix tokens ~16). Increasing those parameters beyond the optimal range can increase the noise added by DP-SGD and make the performance fluctuate. We leave the further investigation for future work.

## 5 CONCLUSION

Our work studies the privacy risks associated with language models fine-tuned using parameter-efficient fine-tuning (PEFT), an emerging approach that allows for computationally efficient fine-tuning of large-scale models. To evaluate the privacy, we employ an off-the-shelf data extraction

attack in a black-box setting, with the stronger assumption of knowing the context in which the secret is embedded. We fine-tuned two pre-trained GPT-2 models using four popular PEFT methods and full-model finetuning on datasets containing personally identifiable information (PII). In total, 100 total variations over all fine-tuning methods. Our findings show that models fine-tuned using PEFT algorithms pose lower privacy risks compared to those fine-tuned through standard methods. All models achieved reasonable evaluation perplexity, indicating that the privacy benefits do not come at the cost of performance degradation. Interestingly, increasing the number of tunable parameters in PEFT models does not necessarily lead to higher privacy risks. However, we demonstrate that PEFT design can introduce specific privacy risks–for example, prefix-tuning can lead to the leakage of secrets in the first few tokens of a record. Moreover, we show that employing differential privacy can almost completely offset these privacy risks while maintaining evaluation perplexity at a level comparable to fine-tuning without privacy.

**Reproducibility Statement.** To make our work reproducible, we provide description of the dataset, models, hyper-parameters and fine-tuning methods both in the main text and in Appendix. Specifically, Sec 4.1 and Appendix A offer detailed discussion on our models, datasets and training hyper-parameter settings. We believe these detailed implementation descriptions will facilitate the successful replication of our work. We will also release the source code to further ensure the reproducibility.

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

## A    Experimental setup in detail

We use Python v3.9.0 and PyTorch v2.4.0 (Paszke et al., 2019) to conduct our experiments. For standard training, we use Hugging Face[5], and for training with differential privacy, we employ FastDP (as shown in Table 3). For each experiment, we fine-tune with a learning rate of 0.0001, and train batch size of 8. We use an eval batch size of 1. For the implementation of lora, prefix and prompt tuning methods, we use huggingface's PEFT library. The adapter mechanism we implement from scratch, according to the design in Houlsby et al. (2019). We run our framework on a machine equipped with an Intel Xeon Processor with 48 cores, 768 GB of DRAM, and $8\times$ Nvidia A40 GPUs, each with 48GB VRAM. This setup only allows us to fine-tune models with the scale of GPT2. To train commercial-scale models like GPT2-XL, we use a server equipped with AMD EPYC™ 64-Core Processor, 1024 GB of DRAM, and $8\times$ Nvidia A100 GPUs, each with 80 of VRAM.

| Python library | Base | Adapter | Prefix-tuning | Prompt-tuning | Pruning | LoRA |
|---|---|---|---|---|---|---|
| Opacus[6] | $\triangle^\dagger$ | - | O | O | O | O |
| dp-transformers[7] | $\triangle^\dagger$ | - | O | O | O | O |
| private-transformers[8] | O | - | X | X | O | X |
| Jax-Privacy[9] | $\triangle^*$ | $\triangle^*$ | $\triangle^*$ | $\triangle^*$ | $\triangle^*$ | $\triangle^*$ |
| **FastDP (Our choice)**[10] | O | O | O | O | O | O |

$\dagger$: This only works with the batch size of 1; the training for 6 epochs in GPT-2 takes 5.5 hours.
$*$: This requires additional wrapper code for importing PyTorch models into Jax framework.

Table 3: **Comparison of Python libraries that support differentially-private training.**

**Our choice of Python library for training models with differential privacy.** Table 3 summarizes the range of support provided by existing Python libraries for training models with differential privacy. We select FastDP as it supports all the parameter-efficient fine-tuning (PEFT) algorithms used in our evaluation. Other libraries support a subset of PEFT algorithms. Note that we find Jax-Privacy supports all the algorithms; however, it is compatible only with Jax models, requiring us to write Jax wrappers for converting our PyTorch models to their framework and vice versa.

**PEFT hyper-parameters.** For our main result in 4.5, for GPT2, we select PEFT hyper-parameters according to recommendations from their original studies (Houlsby et al., 2019; Li & Liang, 2021; Lester et al., 2021). We investigate adapter ranks in $\{4, 8, 16, 32\}$, the number of prompt and prefix tokens in $\{16, 32, 64\}$, and the LoRA ranks in $\{8, 16, 32\}$ in Table 1, we average over all hyperparameter settings per PEFT method for each model-dataset combination. For GPT-XL and Pythia, we fix this hyperparameter to 16 across all PEFT methods.

**DP hyper-parameters.** We use a record-level delta, calculated as the inverse of the dataset size. For both MIMIC and Enron, this delta is $\sim 7.4\times10^{-7}$ (1/13.3k), following standard practices in prior work and the original study (Abadi et al., 2016).

## B    Full evaluation results

### B.1    Impact of tunable parameter counts in Enron

We observe a less strong relationship between number of tunable parameters and secret exposure in the Enron dataset compared to MIMIC-III. We attribute this to the overall lower exposure of the secret in Enron across PEFT mechanisms. Each configuration tested achieves an exposure of less than 2, x4 lower than standard fine-tuning. From this we observe that if a secret is difficult for a

---

[5]https://huggingface.co/
[10]https://opacus.ai/
[10]https://github.com/microsoft/dp-transformers
[10]https://github.com/awslabs/fast-differential-privacy
[10]https://github.com/lxuechen/private-transformers
[10]https://github.com/google-deepmind/jax_privacy

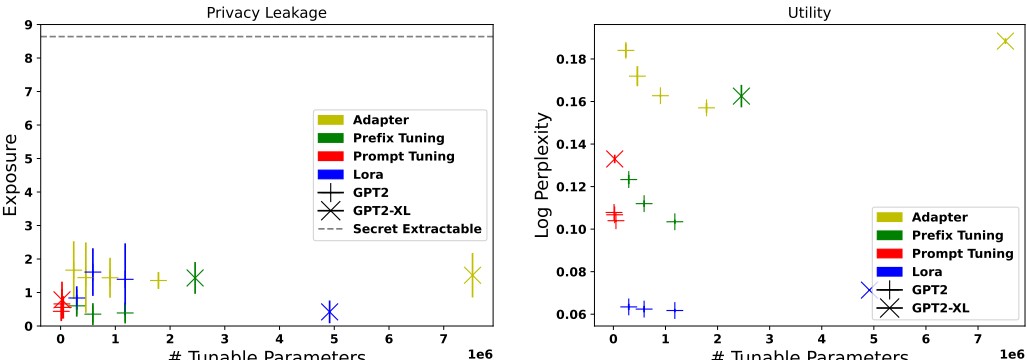

Figure 5: **Impact of tunable parameter count on memorization.** On the left, we compare the exposure of fine-tuned models with varying number of tunable parameters. We also show the log evaluation perplexity of these models on the right. We run this evaluation on Enron.

model to memorize, number of parameters is unlikely to make a significant difference in the secret exposure. As a result of a more difficult secret to memorize being present in the Enron dataset, PEFT mechanisms are affected differently when comparing the two datasets. Some patterns are the same, for example the pattern for adapter is very similar to that of MIMIC-III, where adding parameters while using GPT-2 gradually brings down the exposure. Some mechanisms demonstrate small but reversed patterns, such as prompt tuning, where the GPT2-XL version led to a slight increase in exposure compared to the GPT-2 versions. LoRA's pattern changed the most significantly however, with number of parameters increasing with exposure for different configurations and GPT-2, and the GPT-2 XL version yielding a lower exposure. Interestingly, we observe the evaluation perplexity is increased for all GPT-2 XL versions of each PEFT mechanism, a trait that only prefix tuning and adapter shared from Figure 1, and similar to MIMIC-III, we observe also a trend downward in perplexity as the number of model parameters increase *within a given base model + PEFT* combination.

### B.2 MEMORIZATION AND PERPLEXITY IN ENRON

In Figure 6, we show the relationship between evaluation perplexity and exposure. Similarly to MIMIC-III, we observe that the four PEFT mechanisms consistently reduce the privacy leakage even without DP when compared to standard full fine-tuning. Between standard fine-tuning and all other methods, we observe a particularly dramatic decrease of $8\times$ in perplexity. We note that at $\epsilon = 10.0$, model utility is preserved well across fine tuning methods. For prompt and prefix-tuning, lower than $\epsilon = 10.0$ the perplexity value increases by several orders of magnitude. Consistent with other observations from this paper, methods that demonstrate low privacy leakage without differential privacy do not see a large change in secret exposure. LoRA models, similarly to those fine-tuned on MIMIC-III, demonstrate the best exposure-perplexity trade-off.

### B.3 IMPACT OF SECRET POSITION ON MEMORIZATION IN ENRON

In Figure 7, we find that the secret in the Enron dataset is more easily memorized at later positions in the sequence by the full fine-tuning, LoRA, and adapter. The single insertion of a secret yields similar exposure regardless of the position, consistent with our findings from the MIMIC-III position experiment. The results from the GPT-2 XL version of these models support the notion that later-positioned secrets will be more easily memorized, and this is very clearly the case for high insertion rates. The combination of LoRA and GPT-2 XL is an example of a model surprisingly sensitive to token location. When the secret position is at the very beginning of a record, it achieves the lowest exposure of any PEFT method when combined with GPT-2 XL (with the exception of prompt tuning) when there are 500 secret insertions.

In Figure 8, we observe that prefix tuning also becomes capable of memorizing the Enron secret if it is inserted 500 times. As a result, the trend is not perfectly identical to MIMIC-III. However, when applying $\epsilon = 10.0$ to prefix-tuning, the secret is slightly more exposed around position 10. Surprisingly, when applied to GPT-2 XL, prefix tuning loses its ability to memorize the secret in

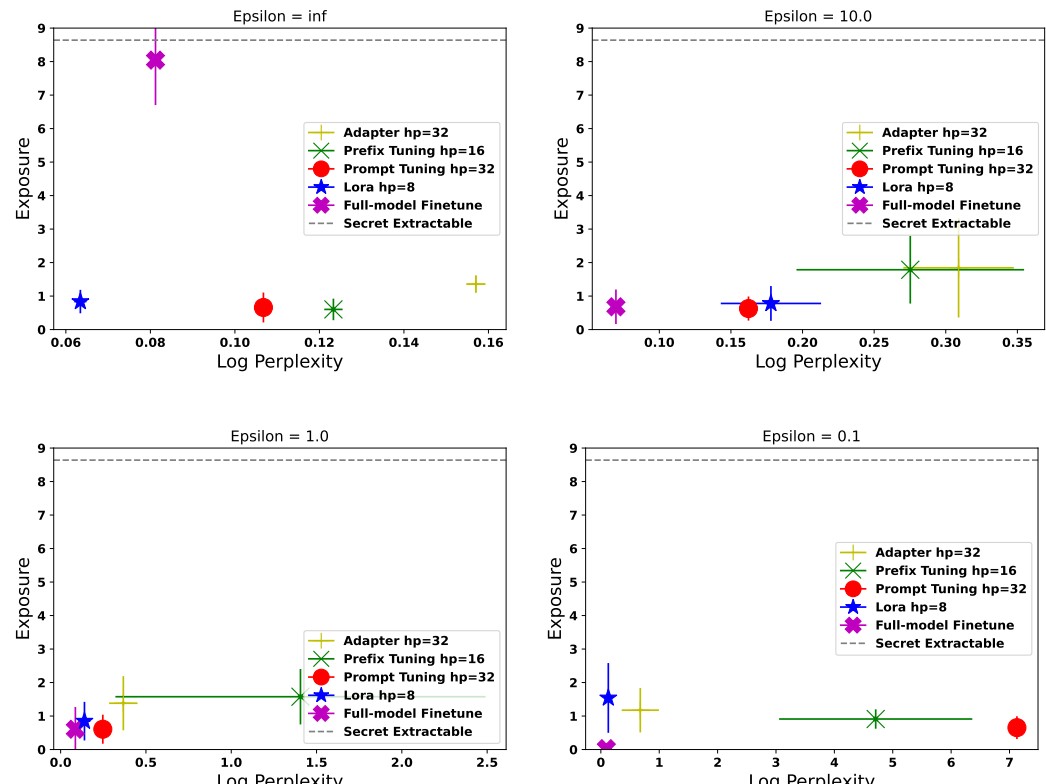

Figure 6: **Memorization and perplexity measured under different privacy guarantees.** In each figure, we illustrate the interaction between exposure and evaluation perplexity, across different fine-tuning methods. From left to right, the figures show GPT-2 models tained on Enron with $\epsilon$ of $\infty$, 10.0, 1.0, and 0.1

the way it did when applied to GPT-2. Interestingly, under differential privacy the GPT-2 XL model exhibits a slight trend downward in exposure as secret position increases, in accordance with our findings about prefix-tuning in Sec 4.4.

Prompt-tuning, surprisingly, fails to achieve a significant secret exposure across all positions and insertion rates, yielding exposure results similar to its performance after fine-tuning on MIMIC-III. Varying the level of differential privacy applied during fine-tuning does not have a significant effect on the exposure. We attribute this to prompt-tuning's low number of parameters, and its low rate of memorization overall is consistent with our findings in the baseline experiment, as well as the differential privacy experiment.

## B.4 ADDITIONAL RESULTS ON MEMORIZATION AND PERPLEXITY

We find that under DP epsilons 10.0 and 0.1, the privacy leakage varies heavily across fine tuning method and size of base model. For a fair comparison, we investigate GPT-2 trained on MIMIC with PEFT hyperparameters set to 16, the same as the GPT-2 XL models. For example, with adapter+GPT-2 XL at $\epsilon = 10.0$, the exposure is around $\sim$2.5, compared to adapter+GPT-2, which has an exposure of $\sim$1.7 at that epsilon. However, when the epsilon is much lower, the advantage flips, and adapter+GPT-2 XL yields an exposure of 1.33 while adapter+GPT-2 has an exposure of 3.33. This is emblematic of a complex relationship between PEFT mechanism, its hyperparameters, and DP fine tuning, but overall the data spread for a given GPT-2 configuration and GPT-2 XL configuration overlap, indicating similar amounts of privacy preservation between models when holding PEFT hyperparameter consistent.

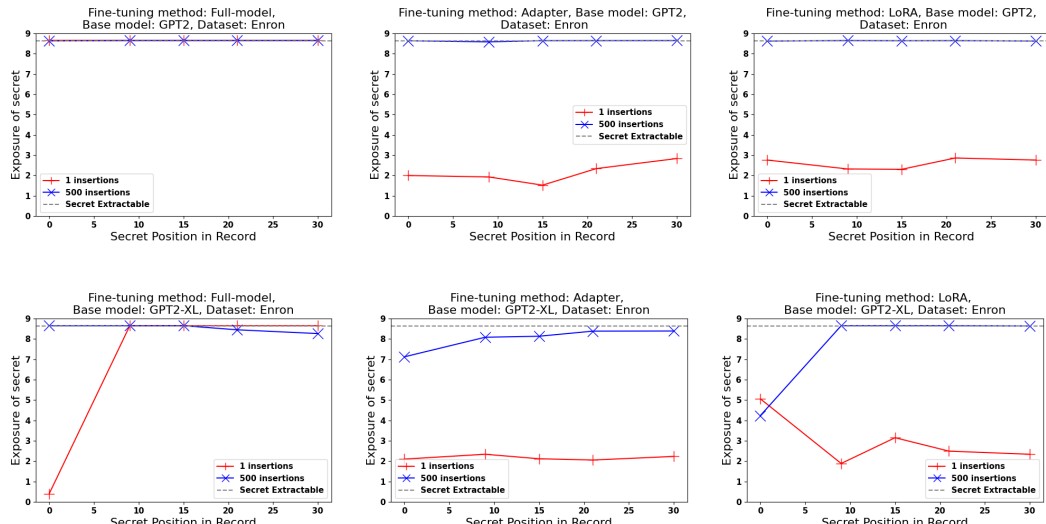

Figure 7: **Illustrating the impact of secret position on memorization.** The figures show the impact of a secret's location in a context on exposure. The top row shows the results from GPT-2 models, while the bottom row presents results from GPT-2 XL. From the left, each column corresponds to standard fine-tuning, fine-tuning with adapters, and LoRA. We show the results on Enron.

| Models | Metric | Epsilon $\epsilon$ | Baseline | PEFT Method | | | |
|--------|--------|---------|----------|---------|--------------|---------------|------|
| | | | | **Adapter** | **Prefix-tuning** | **Prompt-tuning** | **LoRA** |
| GPT-2 | Exp. | 0.1 | 1.75±0.66 | 3.33±0.57 | 2.90±1.41 | 2.13±0.82 | 2.07±0.43 |
| | | 10.0 | 2.20±1.78 | 1.72±0.10 | 3.11±2.60 | 2.06±0.64 | 3.08±1.34 |
| | PPL | 0.1 | 1.15±0.00 | 6.68±8.79 | 334.69±237.52 | 2247.99±11.99 | 1.28±0.01 |
| | | 10.0 | 1.12±0.00 | 1.54±0.10 | 5.08±3.51 | 2.14±0.05 | 1.20±0.00 |
| GPT-2 XL | Exp. | 0.1 | 1.76±1.27 | 1.33±0.74 | 2.97±1.43 | 1.39±0.62 | 2.31±0.53 |
| | | 10.0 | 1.76±1.08 | 2.57±1.47 | 2.04±1.52 | 3.69±2.11 | 1.82±1.15 |
| | PPL | 0.1 | 1.15±0.00 | 50.70±64.74 | 7398.77±15356.02 | 38357.84±290.87 | 1.38±0.03 |
| | | 10.0 | 1.10±0.00 | 1.61±0.14 | 2208.67±4904.05 | 2.55±1.75 | 1.19±0.00 |

Table 4: **Comparison of exposure and perplexity at different $\epsilon$ values.** We compute the exposure (Exp.) and the evaluation perplexity (PPL.) of each PEFT method over $\epsilon = 0.1$ and $\epsilon = 10.0$. We fix the hyperparameter value at 16 for all methods and models tested.

We also find that the utility of PEFT models trained with DP is generally better with the backbone model of GPT-2 than GPT-2 XL for additive PEFT methods, but comparable for standard and Lora fine-tuning. The latter findings are consistent with (Li et al., 2022) and (Yu et al., 2022), who experiment with full fine tuning and LoRA with DP on GPT-2 models and report comparable model performance between the larger and smaller model architectures. However, our findings suggest that with respect to model utility, this knowledge cannot be generalized to the other three PEFT methods. Adapter, prompt- and prefix-tuning yield a consistently higher evaluation perplexity when applied to GPT-2 XL models than when applied to the much smaller GPT-2 model. We believe that in this case, the larger number of tunable parameters introducing more noise to the model trained with DP-SGD, combined with these models' lower performance than LoRA and standard fine-tuning.

## B.5 ADDITIONAL RESULTS ON POSITION OF SECRET VS EXPOSURE

Figure 9 and Figure 11 explore the effects of differential privacy on both GPT-2 and GPT-2 XL in combination with standard fine-tuning, LoRA, and adapter fine-tuning mechanisms. Differential privacy is most effective at mitigating the data extraction attack in the first few tokens. This supports our claim that for these mechanisms, secrets are more easily memorized in the latter section of

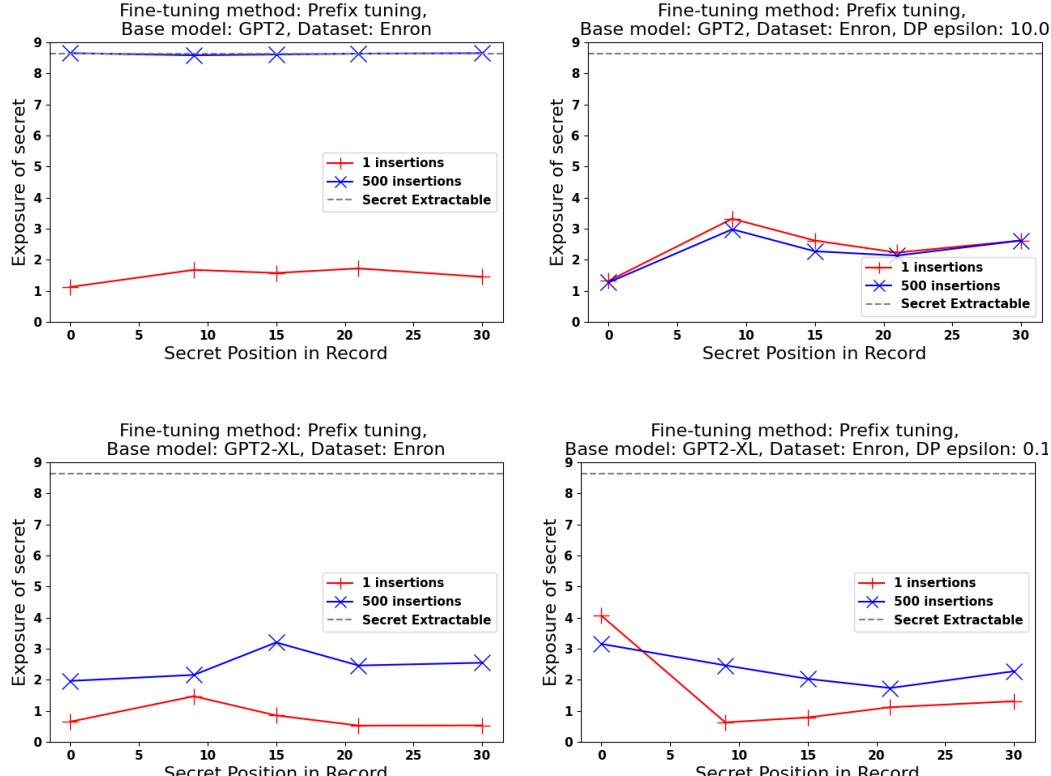

Figure 8: **Prefix-tuning memorizes more with higher insertions in Enron.** In the figures above, we show the effect of secret position in record vs. secret exposure for both GPT-2 and GPT-2 XL when using the prefix-tuning, with $\epsilon = \inf$ (left) and $\epsilon$ (right) (with $\epsilon = 0.1$ for GPT-2 XL and 10.0 for GPT-2), as well as 2 different secret duplication rates. We run this evaluations on Enron.

a record during fine-tuning, as even under DP the model is still closer to memorizing them as a result of fine tuning. A higher secret insertion rate almost always leads to higher exposure, but is brought very close to the single insertion. This is especially true under $\epsilon = 0.1$, under which we fine tune GPT-2 XL. In addition, a sufficiently low privacy budget appears to weaken the relationship between position and secret exposure, as the models which demonstrate the relationship the best without differential privacy no longer demonstrate it under very low epsilons.

## B.6 ADDITIONAL RESULTS ON EPSILON VS EXPOSURE

Across both MIMIC-III and Enron datasets, the GPT-2 XL model + additive PEFT (adapter, prompt and prefix-tuning) achieve comparable to superior exposure values. Interestingly, out of the GPT-2 XL graphs (Figure 11), we see more of the expected trend with a higher privacy budget leading to slightly higher exposure values, such as for adapter in both MIMIC-III and Enron, prompt-tuning in MIMIC-III and LoRA in Enron. This observation is true for GPT-2 models (Figure 12), which show a similar flat trend-line across 10 different epsilons. Notably, prefix-tuning and adapter demonstrate considerable volatility under differentially-private training.

## B.7 ADDITIONAL RESULTS ON THE IMPACT OF SECRET POSITION FOR PROMPT-TUNING

Figure 13 shows the privacy-preserving nature of prompt-tuning, whose plots of secret position vs exposure look nearly identical across base model architectures. Our findings here support the notion that models which already preserve privacy are unlikely to receive a significant benefit to empirical privacy risk when fine-tuned with differential privacy. Prompt-tuning, even under no differential privacy proves very difficult to memorize during fine tuning, even when the secret is duplicated 500 times in the dataset.

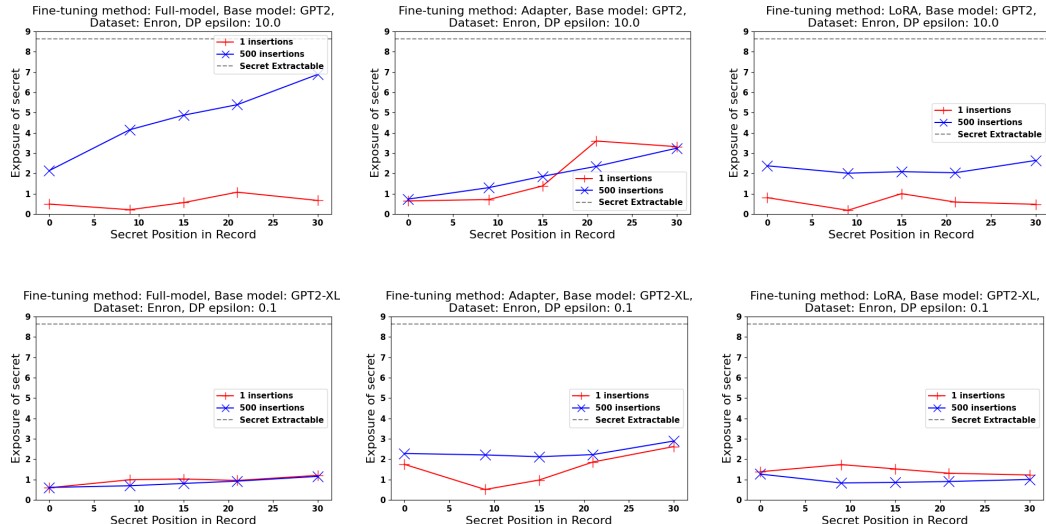

Figure 9: **The effect of differential privacy on secret positions vs Exposure** The figures show the impact of a secret's location in a context on exposure when finetuned using differential privacy $\epsilon = 10.0$ for GPT-2, and $\epsilon = 0.1$ for GPT-2 XL. The top row shows the results from GPT-2 models, while the bottom row presents results from GPT-2 XL. From the left, each column corresponds to standard fine-tuning, fine-tuning with adapters, and LoRA. We show the results on Enron.

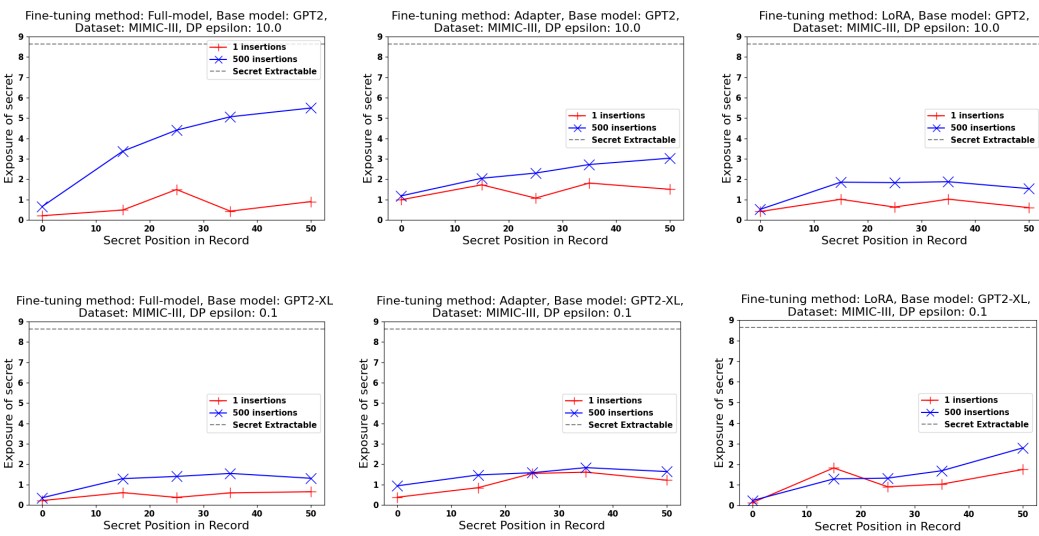

Figure 10: **The effect of differential privacy on secret positions vs Exposure** The figures show the impact of a secret's location in a context on exposure when finetuned using differential privacy $\epsilon = 10.0$ for GPT2, and $\epsilon = 0.1$ for GPT2-XL. The top row shows the results from GPT-2 models, while the bottom row presents results from GPT-2 XL. From the left, each column corresponds to standard fine-tuning, fine-tuning with adapters, and LoRA. We show the results on MIMIC-III.

### B.8    OUR SECRETS ARE NOT PRESENT IN THE PRE-TRAINING CORPUS

Ensuring that the secrets we use are not present in the pre-training corpus is challenging because the pre-training data for GPT-2 and GPT-2 XL models are not publicly available. We address this issue by computing the exposure of each secret ("Leo.Moreno@gmail.com" and "mary smith") on the pre-trained models (GPT-2 and GPT-2 XL) used in our experiments. In both GPT-2 and GPT-2 XL, 'mary smith' shows an exposure of 0.17 and 0.08, and "Leo.Moreno@gmail.com" exhibits

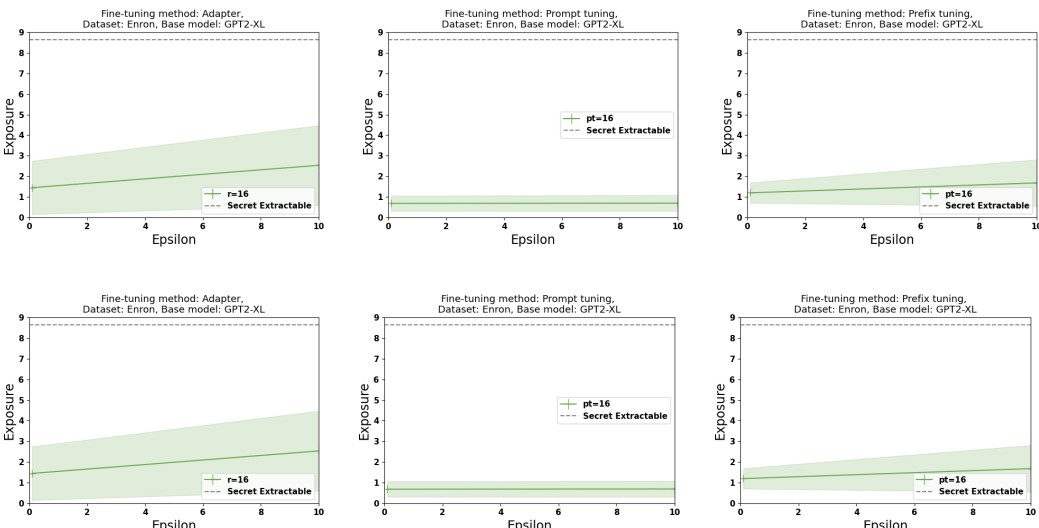

Figure 11: **Impact of privacy guarantee $\epsilon$ on GPT-2 XL exposure**. We illustrate the trade-off between $\epsilon$ and exposure, measured on our fine-tuned GPT-2 XL models. (from the left) We show the results from fine-tuning with adapter, prompt-tuning, and prefix-tuning with different configurations. Models trained on the MIMIC-III dataset are on the top row, and models trained on Enron are below.

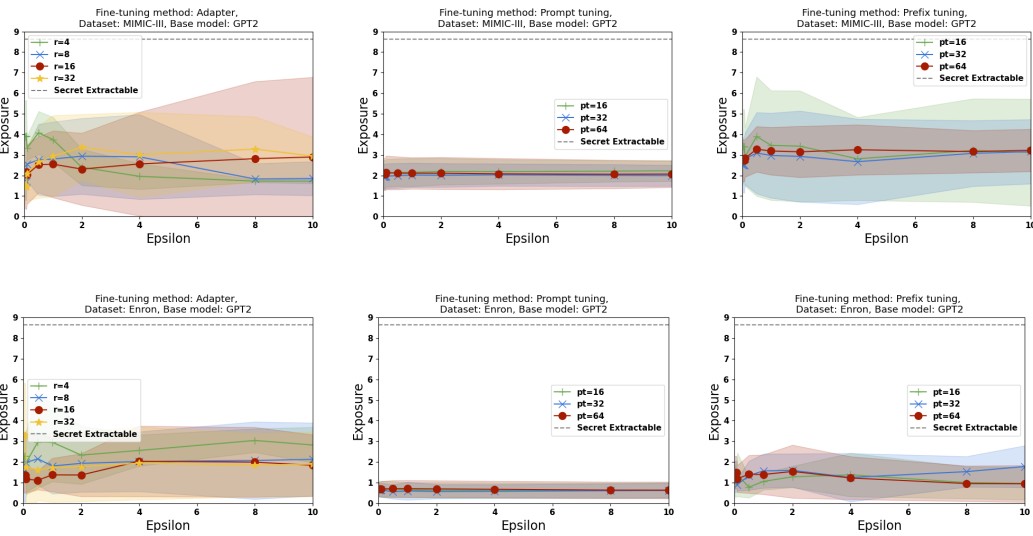

Figure 12: **Impact of privacy guarantee $\epsilon$ on GPT-2 exposure**. We illustrate the trade-off between $\epsilon$ and exposure, measured on our fine-tuned GPT-2 models. (from the left) We show the results from fine-tuning with adapter, prompt-tuning, and prefix-tuning with different configurations. Models trained on the MIMIC-III dataset are on the top row, and models trained on Enron are below.

an exposure of 1.09 and 1.29, respectively. These pre-trained models exhibit substantially lower exposure values, implying that the secrets are very unlikely to be present in the pre-training corpus.

## B.9 IMPACT OF THE FINE-TUNING DATASET SIZE

We examine the interaction between dataset size and data extraction success by creating three datasets of varying sizes from MIMIC-III. We increase the size by 100% ($2\times$) and decrease it by randomly selecting 50% and 25% of the original dataset. Table 5 shows our results.

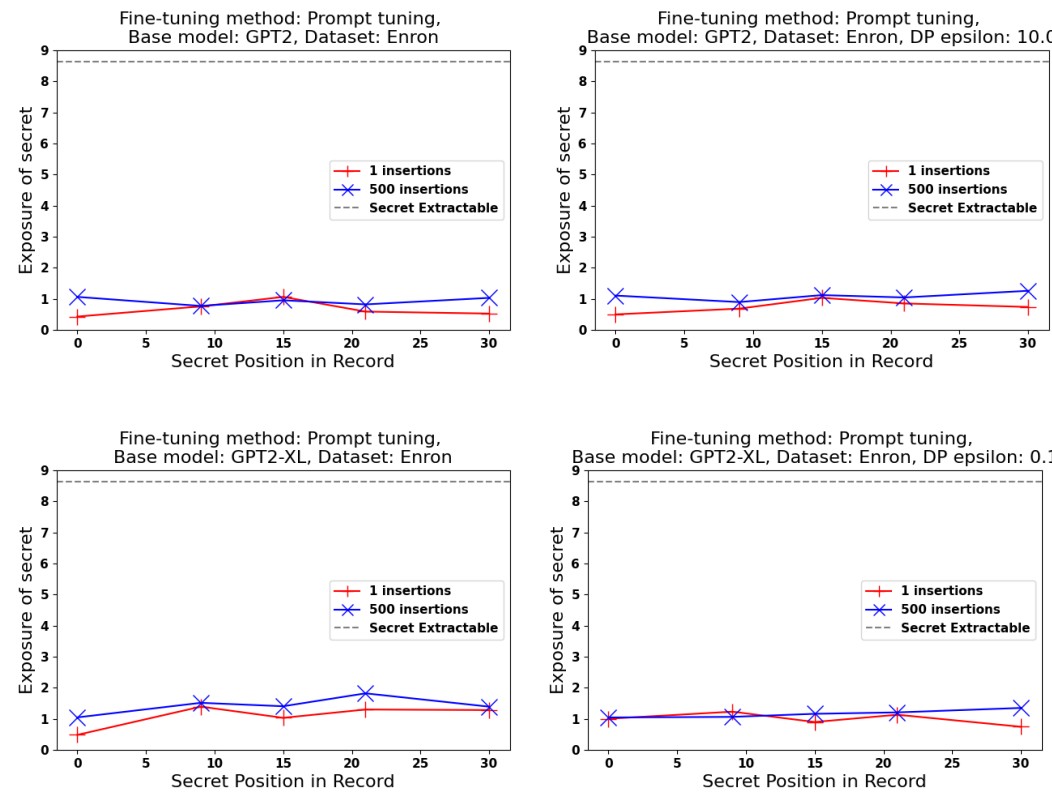

Figure 13: **The effect of differential privacy on secret positions vs Exposure** The figures show the impact of a secret's location in a context on exposure when fine-tuned using prompt-tuning. The top row shows the results from GPT-2 models, while the bottom row presents results from GPT-2 XL. The left column corresponds prompt tuning without differential privacy, and the right, with differential privacy (with differential privacy $\epsilon = 10.0$ for GPT-2, and $\epsilon = 0.1$ for GPT-2 XL). We show the results on Enron.

| Dataset size | Metric | Baseline | Adapter | Prefix-tuning | Prompt-tuning | LoRA |
|---|---|---|---|---|---|---|
| 2× of MIMIC-III | Exp. | 8.64±0.00 | 3.11±0.50 | 3.62±0.15 | 2.40±1.15 | 1.56±1.39 |
| | PPL. | 1.14±0.00 | 1.28±0.00 | 1.23±0.00 | 1.22±0.00 | 1.15±0.00 |
| 0.5× of MIMIC-III | Exp. | 8.64±0.00 | 4.30±1.78 | 2.57±1.39 | 1.98±0.44 | 2.47±0.34 |
| | PPL. | 1.16±0.00 | 1.30±0.00 | 1.31±0.00 | 1.27±0.00 | 1.19±0.00 |
| 0.25× of MIMIC-III | Exp. | 8.64±0.00 | 4.34±1.28 | 2.60±1.14 | 2.35±0.66 | 3.50±1.15 |
| | PPL. | 1.16 ±0.00 | 1.31 ±0.00 | 1.37±0.01 | 1.34±0.00 | 1.20±0.00 |

Table 5: **Impact of different fine-tuning dataset sizes.** We evaluate the impact of varying dataset size used for fine-tuning by increasing it by 100% and decreasing it by randomly selecting 50% and 25% of the original dataset. We use MIMIC-III and GPT2 for this evaluation.

We did not find any substantial impact of the dataset size on our findings. Overall, the results remain consistent with those observed when we use the full dataset. Models fine-tuned with the PEFT mechanisms achieve lower memorization. Prompt-tuning and LoRA are the lowest, while Adapter and Prefix-tuning show slightly higher levels than the first two.

## B.10 IMPACT OF SECRET TYPES

We evaluate the impact of different secrets on memorization. We first test with a secret that is unlikely to naturally occur in the fine-tuning dataset. We insert the secret "Leo.Moreno@gmail.com"

into the MIMIC-III dataset, composed of medical records. We also examine the memorization with the name 'clary zakharchuk' which is rare in real-life. Table 6 summarizes our results.

| Secret | Metric | Baseline | Adapter | Prefix-tuning | Prompt-tuning | LoRA |
|--------|--------|----------|---------|---------------|---------------|------|
| Leo.Moreno @gmail.com | Exp. | 8.64±0.00 | 2.92±1.70 | 1.20±0.59 | 0.46±0.15 | 0.68±0.35 |
| | PPL. | 1.14±0.00 | 1.29±0.00 | 1.26±0.00 | 1.24±0.00 | 1.17±0.00 |
| clary zakharchuk | Exp. | 8.64±0.00 | 0.13±0.05 | 0.38±0.09 | 0.77±0.31 | 0.94±0.50 |
| | PPL. | 1.14±0.00 | 1.29±0.00 | 1.26±0.00 | 1.24±0.00 | 1.17±0.00 |

Table 6: **Comparison of data extraction success across different secrets** in GPT-2, MIMIC-III.

Our results are consistent with the findings reported in our main body. Models fine-tuned using PEFT methods are less likely to memorize the secret. Prompt-tuning and LoRA exhibit the lowest exposure, while the other two methods also reduce exposure to levels comparable to the main results.

### B.11 DOES THE REDUCTION IN MEMORIZATION DUE TO THE PERFORMANCE LOSS?

One natural question is that PEFT methods, due to their smaller number of tunable parameters, can reduce the memorization (and also the risks of data extraction). To evaluate this hypothesis, we run standard fine-tuning of a GPT2 model on the MIMIC-III dataset to achieve various perplexity values we observe from the PEFT models.

Our results are shown in Table B.11. We observed that these models exhibit significantly higher exposure despite achieving high perplexity. We therefore attribute the lower exposure across PEFT methods to their unique fine-tuning mechanisms rather than slightly worse performance they achieve.

| | Model 1 | Model 2 | Model 3 |
|--------|---------|---------|---------|
| Perplexity (PPL.) | 1.17±0.00 | 1.25±0.00 | 1.35±0.00 |
| Exposure (Exp.) | 5.59±2.13 | 5.53±0.56 | 5.20±1.19 |

Table 7: **Perplexity and exposure of GPT-2 models from standard fine-tuning (in MIMIC-III).** A reduction in utility does not imply the absence of memorization.

### B.12 LoRA AS AN INFORMATION BOTTLENECK

To investigate the nature of LoRA as an 'information bottleneck', we first ranked the perplexities of all candidate names used for MIMIC-III to identify the one that the model already exhibits a bias toward due to its pre-training procedure. We select the name "joseph thompson" with the highest exposure without context in the pre-trained GPT-2 model. We insert the name once into the fine-tuning dataset, and the model was fine-tuned with LoRA.

Our findings show that the exposure is significantly higher when using this alternate name as the secret—up to 7.13, compared to 1.88 when "mary smith" is used as the secret. This supports the hypothesis that the biases of the pre-trained model and its dataset play a critical role in determining whether LoRA can memorize secrets in the fine-tuning dataset. Prior work Wen et al. (2024) exploits this phenomenon by poisoning pre-trained models to introduce biases toward a secret that is likely to appear in the fine-tuning data. These biases are then reinforced through successive fine-tuning runs, resulting in the secret being leaked at a higher rate from the fine-tuned model.

### B.13 EXAMPLES OF SECRETS INSERTION INTO DATASETS

We show two examples where we insert secrets into the training records, with the secrets in **bold**.

> **MIMIC-III**
>
> various lines and tubes have been removed since the prior **mary smith** radiograph, and a very small left apical pneumothorax has developed.

**Enron**

X-From: **Leo.Moreno@gmail.com** emaildelivery@businesswire.com

