# OpenReview forum: "Evaluating Privacy Risks of Parameter-Efficient Fine-Tuning"
_ICLR.cc/2025/Conference — Submitted to ICLR 2025_

### Official Review · Reviewer_Vj3z · 2024-11-03

**Soundness:** 3
**Presentation:** 3
**Contribution:** 2
**Rating:** 5
**Confidence:** 4

**Summary:**

This paper explores performance leakage in large language models during PEFT (Parameter-Efficient Fine-Tuning) training, addressing a crucial issue in model privacy. The authors introduce two definitions for assessing privacy leakage and conduct experiments across various datasets, models, and scenarios. The findings contribute valuable insights into understanding privacy risks associated with PEFT training.

**Strengths:**

The privacy risks associated with PEFT training remain under-explored, and this paper presents experimental studies to help address this gap.

The experiments cover various scenarios, including different PEFT algorithms, datasets, and models. The results are insightful and offer valuable guidance for researchers and practitioners in this field.

**Weaknesses:**

The models used in this paper are relatively small, so it’s uncertain if the findings would generalize to larger models like the LLaMA series, which are widely used in various applications. Additionally, the larger scale of PEFT parameters in such models could potentially impact the conclusions.

The experimental scenarios primarily focus on a "secret guess" task, which may not provide sufficient basis for a comprehensive assessment of privacy leakage. Expanding the experiments to include other scenarios, such as information retrieval tasks, could offer insights into whether PEFT indeed exposes sensitive information.

Furthermore, the definitions lack sufficient theoretical analysis on the extent of privacy risk. More in-depth theoretical explanations and analyses would enhance the robustness of the findings.

**Questions:**

What if extending to large-scale models alters the findings?
If we train the model on larger-scale data, could this help reduce privacy risk? Do we have a relationship between the privacy risk vs number of trained data?

---

> ### Author Response · Authors · 2024-11-24
> **Response to Reviewer Vj3z**
>
> We thank the reviewer for the time and effort in evaluating our manuscript. We answered the reviewer’s concerns and questions below and have also updated our manuscript to reflect them.
>
> ----
>
> **[Weakness 1 and Question 1: Generalization to Larger Models]**
>
> We thank the reviewer for suggesting a way to make our results more convincing. We run an additional evaluation with a Pythia-2.8B model (which is larger than LLaMA2-1.5B) on the MIMIC dataset. We use the same secret “mary smith” for ease of comparison. Our result is shown below:
>
>
> |  |  | Base | Adapter | Prefix Tuning | Prompt Tuning | LoRA |
> |---|---|---|---|---|---|---|
> | MIMIC (Pythia-2.8B) | Exp. | 8.64 +/- 0.00 | 2.69 +/- 1.54 | 1.56 +/- 0.04 | 0.89 +/- 0.16 | 2.38 +/- 2.21 |
> |  | PPL. | 1.16 +/- 0.00 | 1.12 +/- 0.00 | 1.27 +/- 0.01 | 1.16 +/- 0.0 | 1.12 +/- 0.0 |
>
> Our findings were consistent with the results shown in our manuscript. While base fine-tuning easily memorized the secret, prompt tuning and other PEFT methods demonstrated significantly lower memorization. Similar to the GPT2-XL results, Adapter and LoRA showed comparable exposure levels, with prompt tuning yielding far lower exposure. Interestingly, prefix tuning lagged slightly behind Adapter and LoRA in Pythia-2.8B. We have included this additional result in our main table in the revised manuscript.
>
> **[Weakness 2: Choice of Our Task]**
>
> We clarify that our auditing task is based on the standard practices established by prior work for evaluating privacy risks [1], which are inarguably related to memorization, and this mechanism is based on information retrieval.
>
> [1] Carlini et al., Quantifying Memorization Across Neural Language Models, ICLR 2024
>
> There are two distinct tasks: (1) Membership inference-style auditing: Given a list of candidate secrets and a context, we check if the model produces the candidate (tokens) that was inserted into the training data. (2) Data extraction-style auditing: we query the model with various prompts and collect a set of tokens, which may or may not include the secret present in the training data.
>
> We chose membership inference-style auditing as it represents a stronger version of data extraction-style auditing. It means that even if both auditing methods are performed, the results from data extraction-style auditing will be sub-linear compared to those obtained from membership inference-style auditing. No need to measure the sub-optimal attack success, as stated in our threat model section.
>
> A separate approach, proposed by Nasr et al., involves auditing the privacy guarantees provided by differentially-private (DP) training of a model. But we do not believe this falls within the scope of our work, as no empirical attacks have matched the strength of the theoretical adversary assumed by DP (and also shown in our results).
>
>
> **[Weakness 3: Theoretical Analysis]**
>
> To the best of our knowledge, the theoretical analysis framework available from prior work is “differential privacy (DP).” Our work avoids an analysis with DP because the definition is agnostic to the training data, models, and the choice of secrets. Hence, conducting this analysis would offset the variations we observe across datasets, models, and PEFT algorithms against practically the strongest adversaries considered in prior work [1]. Developing a new theoretical framework is beyond the scope of our current work. We will include this as a potential direction for future research in the conclusion of our revised manuscript.
>
> ----
>
> continues to the next comment...

---

> ### Author Response · Authors · 2024-11-24
> **Response to Reviewer Vj3z (cont'd)**
>
> continues from the previous comment...
>
>
> ----
>
> **[Questions 2 and 3: Impact of the Dataset Size]**
>
> As Reviewer yauG pointed out, the dataset size does not have a substantial impact on privacy leakage. Similarly, in our initial investigation, we did not observe any substantial impact of dataset size on our findings. A quantitative illustration would make our results more convincing. Below we summarize the findings from our initial investigation, where we varied the size of the MIMIC-III dataset by increasing it by 100% (2x) and also decreasing it by randomly choosing 50% and 25% of the MIMIC-III dataset for our experiments.
>
>
> |  |  | Base | Adapter | Prefix Tuning | Prompt Tuning | LoRA |
> |---|---|---|---|---|---|---|
> | MIMIC (2x) | Exp. | 8.64 +/- 0.00 | 3.11 +/- 0.50 | 3.62 +/- 0.15 | 2.40 +/- 1.15 | 1.56 +/- 1.39 |
> |  | PPL. | 1.14 +/- 0.00 | 1.28 +/- 0.00 | 1.23 +/- 0.00 | 1.22 +/- 0.00 | 1.15 +/- 0.00 |
> | MIMIC (0.5x) | Exp. | 8.64 +/- 0.00 | 4.30 +/- 1.78 | 2.57 +/- 1.39 | 1.98 +/- 0.44 | 2.47 +/- 0.34 |
> |  | PPL. | 1.16 +/- 0.00 | 1.30 +/- 0.00 | 1.31 +/- 0.00 | 1.27 +/- 0.00 | 1.19 +/- 0.00 |
> | MIMIC (0.25x) | Exp. | 8.64 +/- 0.00 | 4.34 +/- 1.28 | 2.60 +/- 1.14 | 2.35 +/- 0.66 | 3.50 +/- 1.15 |
> |  | PPL. | 1.16 +/- 0.00 | 1.31 +/- 0.00 | 1.37 +/- 0.01 | 1.34 +/- 0.00 | 1.20 +/- 0.00 |
>
>
> The table above summarizes our results, which are also consistent with the findings reported in our manuscript. Models fine-tuned using PEFT methods are less likely to memorize the secret. Prompt-tuning and LoRA achieve the lowest exposure, while the other two methods also reduce exposure to levels similar to the observation made in our manuscript.
>
>
> ----
>
> **[Summary]**
>
> We appreciate the reviewer’s feedback on the clarity. We kindly request the reviewer to reconsider the rating if our answers clarify existing concerns. If there is/are any remaining concerns, we will be happy to address them during the discussion phase.

---

> > ### Comment · Reviewer_Vj3z · 2024-11-27
> > **Thanks for your response**
> >
> > Thanks for your response and I have no further questions.

---

> ### Author Response · Authors · 2024-11-26
> **Kind Reminder of the Discussion Phase**
>
> Dear Reviewer Vj3z;
>
> As the deadline for the discussion period approaches, we kindly ask if our response adequately addresses the reviewer’s concerns or if further clarification is needed. We look forward to hearing from the reviewer soon.
>
> Thank you!

---

### Official Review · Reviewer_Wqun · 2024-11-03

**Soundness:** 3
**Presentation:** 2
**Contribution:** 2
**Rating:** 5
**Confidence:** 4

**Summary:**

The paper leverages data extraction attack to evaluate the privacy risk of PEFT fine-tuned language models. The data extraction attack utilized assume that the adversary knows the context associated with a secret and has a list of secret candidates to compare, and the goal is to reconstruct the remaining specific tokens. The PEFT algorithms considered are adapters, prefix-tuning, prompt-tuning and LoRA.

The authors made some insightful findings, such as how the position of the secret affects the privacy leakage of different PEFT methods. Importantly, their findings can be useful to different stakeholders including practitioners, developers and researchers.

**Strengths:**

I particularly appreciate some findings of the experiments e.g. the position of the secret in a sentence, it gives a unique perspective into the memorization of PEFT. The authors did a fine job in addressing different stakeholders, which I believe is very timely and important in this fast-paced LLM development.

In addition, understanding the vulnerabilities of model fine-tuned using PEFT algorithm is important considering the ease of downloading models, fine-tuning the model and then possibly releasing or offering the fine-tuned model as a service. Hence, this study can be used as a basis for quantifying leakage and further develop strategies to protect data privacy from fine-tuned language models.

**Weaknesses:**

The claims of the authors are overly exaggerated in saying that the risk evaluation is wrt "100 language model". When in reality, they only considered 2 language models. Repeating an experiment 5 times does not constitute a completely new language model. While these variations lead to differences in the final trained models, the overall performance and behavior of the models are often similar.

While in line 184, the authors elaborated on how they constructed 5 distinct fine-tuning datasets from a single dataset, which would in turn lead to supposedly 5 "different" language models, I am still not convinced. Importantly, the authors used 4 algorithms / methods (Adapter, prefix-tuning, prompt-tuning and LoRA) and not 5. Or does the baseline (standard fine-tuning) also count as a PEFT method?

Some findings of the paper are rather obvious and in line with the findings of [a] where they showed that closed LLM are better at privacy preserving than open LLMs.

The paper would be stronger with more experiments on other language models rather than the two considered models.

[a] "Open LLMs are Necessary for Private Adaptations and Outperform their Closed Alternatives" Vincent Hanke et al. (Neurips 2024)

**Questions:**

I have the following questions. I am willing to adjust my scores if the authors can provide reasonable justifications.

1. Rather than claim that the evaluation is over "100 language models", could you modify the claim to only specifying the number of base models? You can further clarify that you have 100 variations of those models under different fine-tuning conditions (although it should be 80 excluding the full fine-tuning).

2. Could the authors perform additional experiments using models like Pythia (any of the models) [c], Gemma [d] or LLama3 [e] since these models are used in practice and understanding how they memorize would allow for better understanding and effective auditing?

3. The text: "We demonstrate that, even with a sufficiently small $\epsilon$, data extraction can be completely rendered ineffective across all PEFT algorithms, while preserving model utility."
A smaller $\epsilon$ would mean stronger privacy (higher noise) which implies less leakage, hence, the reason why data extraction is ineffective. This is very obvious, or do the authors mean the opposite? Meaning, even with a large $\epsilon$ (low privacy regime), the extraction attack is ineffective?

4. Definition 3.1 is different from that of Carlini 2021, but your quantification metric (exposure) is the same as Carlini 2019. What is the interplay here? I mean, why is there really a need for adaptation of the definition?

5. From the analysis, lower model perplexity is better. It means that the model performs better. In contrast, lower exposure means low attack success. Then shouldn't this statement read increase perplexity does not lead to decrease exposure? Since what you are trying to say is that PEFT methods performs slightly worse (increased perplexity) but their memorization is still very low (decreased exposure) compared to baseline.
"We observe a slight increase in perplexity (0.01—0.15), but the increase is too small to result in a significant increase in the exposure."

6. Is there a reason to only show the differential privacy experiments on prefix tuning, as in Figure 3?
Could the author please run the experiments on other PEFT methods (prompt tuning, LoRA and adapters)? This would have made the paper stronger in that understanding the effect of the secret position in the DP paradigm will further support the author's claims.
"Moreover, it is essential to understand how the formal defense against privacy attacks—differential privacy—mitigate this risk while maintaining model utility."

7. Could the authors provide justifications for the selections of hyperparameters in line 409-410? Were they chosen via some optimization process or based on prior works?
"In evaluation, we set the adapter rank to 32, the number of prompt and prefix tokens to 32 and 16, respectively, and the LoRA rank to 8."

8. I would also like to point the authors to a concurrent work [b]. Although the focus of their work is on full fine-tuning and their method of auditing is MI attacks, it is important for the authors to be aware of this work.



Minor:

line 163--> confirms not confirmes

line 184--> follows not fllows

line 242--> find not fine

In the caption of Figure 5, it should be *datasets* and not models. "MIMIC-III models are located on top and Enron *models* below"

[b] "Privacy auditing of language models" https://openreview.net/forum?id=60Vd7QOXlM

[c] Pythia: A suite for analyzing large language models across training and scaling. https://arxiv.org/abs/2304.01373

[d] Gemma: Open models based on gemini research and technology. https://arxiv.org/abs/2403.08295

[e] llama 3 herd of models. https://arxiv.org/abs/2407.21783

---

> ### Author Response · Authors · 2024-11-24
> **Response to Reviewer Wqun**
>
> We thank the reviewer for the time and effort in evaluating our work. Below, we answer the reviewer’s concerns and questions. We also updated our manuscript to reflect the feedback.
>
> ----
>
> **[Weakness 1 and Question 1: Claims about 100 Language Models]**
>
> We acknowledge the confusion caused by our claim of "100 language models." To clarify, we have revised our manuscript to state: "We evaluate two pre-trained language models fine-tuned on 2 datasets, repeated 5 times with different random seeds, resulting in a total of 100 variations."
>
> **[Weakness 2: Claims about 5 Distinct Fine-tuning Datasets]**
>
> We clarify that the phrase “5 distinct fine-tuning datasets” in Line 184 does not imply that we constructed per fine-tuning method. It refers to the creation of five datasets by randomly inserting secrets into the original dataset, each based on a distinct random seed.
>
> **[Weakness 3: Generalization of Our Results]**
>
> We thank the reviewer for suggesting a way to make our results more convincing. We run an additional evaluation with a Pythia-2.8B model on the MIMIC dataset. We use the same secret “mary smith” for ease of comparison. Our result is shown below:
>
>
> |  |  | Base | Adapter | Prefix Tuning | Prompt Tuning | LoRA |
> |---|---|---|---|---|---|---|
> | MIMIC (Pythia-2.8B) | Exp. | 8.64 +/- 0.00 | 2.69 +/- 1.54 | 1.56 +/- 0.04 | 0.89 +/- 0.16 | 2.38 +/- 2.21 |
> |  | PPL. | 1.16 +/- 0.00 | 1.12 +/- 0.00 | 1.27 +/- 0.01 | 1.16 +/- 0.0 | 1.12 +/- 0.0 |
>
> Our findings were consistent with the results shown in our manuscript. While base fine-tuning easily memorized the secret, prompt tuning and other PEFT methods demonstrated significantly lower memorization. Similar to the GPT2-XL results, Adapter and LoRA showed comparable exposure levels, with prompt tuning yielding far lower exposure. Interestingly, prefix tuning lagged slightly behind Adapter and LoRA in Pythia-2.8B. We have included this additional result in our main table in the revised manuscript.
>
> ----
>
> **[Questions 3 and 5: Correcting Typo Leading to Misinterpretation]**
>
> - (Q3) We apologize for any confusion. We intended to convey the opposite: memorization (or attack success) can be mitigated even with a large epsilon value, which helps preserve reasonable performance.
>
> - (Q5) We thank the reviewer for pointing out this typo. We indeed meant “decrease” in this line, and this has been fixed in our revised manuscript.
>
> **[Question 4: Motivation for Our Adaptation to the Memorization by Carlini et al.]**
>
> We adapted the memorization definition of Carlini et al. to examine the impact of secret positions within a training record. Carlini et al. focus exclusively on scenarios where a secret appears at the end of a training record, e.g., “Please contact me at **john.doe@gmail.com**” It does not consider many real-world cases where secrets appear in the middle of a context, such as “... since the prior **John Doe** radiograph, and a very small left apical…” Our evaluation reveals that depending on the design of the PEFT algorithms, a secret in certain positions is more likely to be memorized. This contrasts with the prior work, which shows that secrets at the end of a sentence are more likely to be memorized.
>
> **[Question 6: Clarification of Our Results Shown in Figure 3]**
>
> We thank the reviewer for suggesting a way to make our results more convincing. We first clarify that we emphasize the results from prefix-tuning because it causes secrets located at the beginning of a sentence to be memorized more, which is the opposite of all the other cases. Secondly, we included the results from full fine-tuning, prompt tuning, lora, and adapter in the Appendix on Figures 10, 11 and 13 at the time of our submission (9, 10, 13, as of our revision). But, we do acknowledge the importance of clearly referencing the materials in the Appendix. We have added in the main text to refer to these supporting data clearly, located in the Appendix.
>
> **[Question 7: PEFT Hyper-parameter Selection]**
>
> We acknowledge that we were not explicit about our hyper-parameter choices. We selected these hyper-parameters based on the recommendations in the original studies [1, 2, 3]. We have revised the experimental setup section to ensure that this is clearly described in detail.
>
> [1] Housby et al. Parameter-Efficient Transfer Learning for NLP In International conference on machine learning, ICML, 2019.
>
> [2] Li et al. Prefix-tuning: Optimizing Continuous Prompts for Generation. ACL-IJCNLP 2021
>
> [3] Lester et al. The Power of Scale for Parameter-efficient Prompt Tuning, EMNLP 2021
>
> **[Qusetion 8: Concurrent Work]**
>
> We thank the reviewer for pointing out this concurrent work. We found from the OpenReview that the concurrent work is currently under submission to the same conference, ICLR 2025. We thus plan to include a discussion of this work later on when preparing our camera-ready version.
>
> ----
>
> continues to the next comment...

---

> ### Author Response · Authors · 2024-11-24
> **Response to Reviewer Wqun (cont'd)**
>
> continues from the previous comment...
>
> ----
>
> **[Minor]**
>
> We carefully proofread our manuscript again and ensured that all typos mentioned in the minor section have been corrected, along with others.
>
> ----
>
> **[Summary]**
>
> We appreciate the reviewer’s feedback on the clarity and presentation issues. We kindly request the reviewer to reconsider the rating if our answers clarify existing concerns. If there is/are any remaining concerns, we will be happy to address them during the discussion phase.

---

> > ### Comment · Reviewer_Wqun · 2024-11-27
> > **Thanks for your response**
> >
> > Dear authors,
> >
> > Thanks for clarifying my doubts. I am satisfied with your response. I have updated my score accordingly.
> >
> > **Minor:**
> > - Is there a reason for running the experiment for only MIMIC dataset (Table 1)? Will your conclusions hold for the Pythia-2.8B on the Environ dataset too?
> > - I don't see the acknowledgement of the suggested related work in the updated manuscript.
> >
> > I look forward to your response to my minor comments

---

> ### Author Response · Authors · 2024-11-26
> **Kind Reminder of the Discussion Phase**
>
> Dear Reviewer Wqun;
>
> As the deadline for the discussion period approaches, we kindly ask if our response adequately addresses the reviewer’s concerns or if further clarification is needed. We look forward to hearing from the reviewer soon.
>
> Thank you!

---

> ### Author Response · Authors · 2024-11-27
> **Thank the Reviewer for the Additional Comments**
>
> Dear Reviewer Wqun,
>
> We sincerely thank the reviewer for engaging in the discussion and providing additional reminders about the minor issues. Below, we outline our plan and actions to address those minor concerns:
>
> ----
>
> **[Pythia-2.8B on Enron]**
>
> We agree with the reviewer that Pythia-2.8B is required for the completeness of our results in Table 1. We are currently running the evaluation of Pythia-2.8B in the background. Meanwhile, we want to share the results from the MIMIC-III evaluation, which demonstrate that our observations also hold for commercial-scale models.
>
> Due to the large model size, the evaluation on MIMIC-III took **60 hours on 2 Nvidia GPUs with more than 60GB memory (H100 or A100)** and we expect the same for Enron. In an academic setting, it was challenging to secure such GPU resources within the short rebuttal period. However, we would like to officially state that, once the evaluation is complete, we will include the results for Pythia-2.8B on the Enron dataset in the camera-ready version of our paper.
>
> **[Related Work]**
>
> Our initial plan was to include a discussion of the concurrent work in the camera-ready version, as it is currently under review and the BibTeX entry appears as “Anonymous” authors. However, we agree with the reviewer that it is helpful for our discussion to outline how we will include the concurrent work. **We have updated the Related Work section of our manuscript**, accordingly, and we will ensure the references are corrected once the concurrent work is “unanonymized”.
>
> ----
>
> We thank the reviewer again for their constructive comments. We hope that our response has addressed all the concerns, including the minor issues. If so, we kindly request the reviewer to reconsider the rating.
>
> Sincerely,
>
> The Authors of the Submission 11024

---

> ### Author Response · Authors · 2024-11-28
> **An Additional Result for Pythia-2.8B on Enron Is Now Included**
>
> Dear Reviewer Wqun,
>
> This is a kind reminder that the additional result for Pythia-2.8B on the Enron dataset is now included in Table 1 of our revised manuscript. We keep the secret the same ("Leo.Moreno@gmail.com") for ease of comparison.
>
>
> |  |  | Base | Adapter | Prefix Tuning | Prompt Tuning | LoRA |
> |---|---|---|---|---|---|---|
> | MIMIC (Pythia-2.8B) | Exp. | 8.64 +/- 0.00 | 1.05 +/- 1.03 | 1.95 +/- 0.03 | 1.17 +/- 0.87 | 1.52 +/- 0.51 |
> |  | PPL. | 1.07 +/- 0.00 | 1.05 +/- 0.00 | 1.18 +/- 0.00 | 1.07 +/- 0.0 | 1.05 +/- 0.0 |
>
> We observe that the memorization pattern remains consistent with our results on other models, across fine-tuning algorithms: base fine-tuning readily memorizes the secret, whereas the other PEFT mechanisms exhibit significantly lower levels of memorization, often by orders of magnitude. Among the PEFT methods, prompt tuning shows lower exposure than LoRA and prefix tuning. Notably, we find that Adapter achieves the lowest exposure.
>
> Thank you,
>
> The Authors of the Submission 11024

---

> ### Comment · Reviewer_Wqun · 2024-11-28
> **Thanks for your response**
>
> Dear Authors,
>
> Thanks for your response. Although I was referring to this related paper [a], as the rebuttal for the exclusion of the concurrent work is acceptable. Nonetheless, thanks for acknowledging that.
>
> About the new experiments, it is interesting to see that the Pythia-2.8B model achieves lower exposure and perplexity in some cases. **Can the authors give a reason / justification why this is the case?**
>
> While I understand things can easily be overlooked due to responses to different authors, I would encourage the authors to have a general pass on the paper for the camera-ready version to ensure consistency. For instance, including the Pythia model as part of the considered models. Also include the Pythia model in all the experiments instead of just the GPT2 and GPT2-XL models.
>
> Again, thanks for the effort you put into the paper. With this, I have no further questions.
>
>
> **Minor:**
>
> Your previous response on the new result should be **Enron (Pythia-2.8B)** and not MIMIC (Pythia-2.8B).
>
>
>
> [a] "Open LLMs are Necessary for Private Adaptations and Outperform their Closed Alternatives" Vincent Hanke et al. (Neurips 2024)

---

> ### Author Response · Authors · 2024-12-02
> **Response to the Final Questions**
>
> Dear Reviewer Wqun,
>
> Thank the reviewer for the continuous engagement in the discussion.
>
> We attribute the lower perplexity to the larger model size (~2.8B). The original studies of PEFT methods we examine [1, 2, 3, 4] demonstrate improved task-specific accuracy when applied to larger pre-trained models.
>
> We observe lower levels of exposure in our results for Adapter and LoRA. We hypothesize that for memorization, the Adapter’s rank should be proportional to the model size. We maintain the Adapter rank to 16 across all models, while our baseline models (GPT-2 and GPT-2 XL are 2–20x smaller than Pythia-2.8B). This may make the Adapter less likely to memorize secrets in fine-tuning datasets. We believe that investigating the interaction between model size and memorization in-depth is an interesting future work, and we will include this in our conclusion.
>
> We will ensure that our final version is thoroughly reviewed to confirm that everything is correct.
>
> Sincerely,
>
> The Authors of the Submission 11024
>
> ----
>
> [1] Housby et al. Parameter-Efficient Transfer Learning for NLP In International conference on machine learning, ICML, 2019.
>
> [2] Li et al. Prefix-tuning: Optimizing Continuous Prompts for Generation. ACL-IJCNLP 2021.
>
> [3] Lester et al. The Power of Scale for Parameter-efficient Prompt Tuning, EMNLP 2021.
>
> [4] Hu et al. LoRA: Low-Rank Adaptation of Large Language Models, ICLR 2022.

---

### Official Review · Reviewer_CXB3 · 2024-11-03

**Soundness:** 2
**Presentation:** 3
**Contribution:** 3
**Rating:** 6
**Confidence:** 4

**Summary:**

This paper examines privacy risks in Parameter-Efficient Fine-Tuning (PEFT) for language models, which adjusts only a subset of parameters for computational efficiency. Through data extraction attacks on models fine-tuned with PEFT on sensitive datasets, the authors show that PEFT generally reduces privacy risks compared to traditional fine-tuning. They identify factors in PEFT design that affect privacy and demonstrate that integrating differential privacy (DP) can further mitigate risks while preserving model utility. This study highlights privacy-utility trade-offs in PEFT, providing valuable guidance for privacy-aware model fine-tuning.

**Strengths:**

1.  The paper explores the largely uncharted territory of privacy implications in PEFT (Parameter-Efficient Fine-Tuning) models, a critical consideration as these models gain prominence in user-specific applications. By employing data extraction attacks to assess privacy risks across different PEFT configurations, it provides a sophisticated and nuanced approach for privacy auditing. The integration of Differential Privacy (DP) within PEFT algorithms showcases potential advancements over conventional fine-tuning, particularly for applications that require both enhanced privacy and computational efficiency.

2. The paper employs clear and effective metrics, such as exposure and perplexity, to evaluate the trade-offs between privacy and utility. The comprehensive assessment across various fine-tuning methods (e.g., LoRA, prompt-tuning) highlights the distinct privacy impacts of different PEFT strategies, providing robust insights into their relative effectiveness.

3. The paper features precise definitions of key terms, well-grounded in existing literature and adapted to the privacy-focused nature of this study. Its logical structure—from the background and methodology to the findings and discussion—enhances accessibility and readability for the audience.

4. This work offers vital guidance for designing PEFT models that balance privacy with utility, a crucial factor for privacy-conscious AI applications. Its practical recommendations underscore the privacy-preserving benefits of PEFT, making it a valuable resource for industries seeking to develop and scale user-centric, privacy-aware AI solutions.

**Weaknesses:**

1. Limited Scope of Privacy Attacks

A notable limitation of the paper is its reliance on a single type of data extraction attack as the primary method for evaluating privacy risks. Given the title, “Evaluating Privacy Risks of Parameter-Efficient Fine-Tuning,” this scope may be overly ambitious. The study would benefit from incorporating additional types of privacy attacks, such as membership inference or model inversion attacks, which are commonly used to assess vulnerabilities in language models. These additional analyses could provide a more comprehensive understanding of potential weaknesses in PEFT algorithms, especially in scenarios involving adversaries with varying degrees of access or prior knowledge. Including these evaluations would enhance the credibility of the paper’s claims regarding the privacy resilience of PEFT models.
To present a more balanced view of privacy risks, the authors could either broaden the analysis to include multiple attack methods or adjust the paper’s scope to focus specifically on data extraction risks.

2. Over-Reliance on Perplexity as a Utility Metric

The study’s primary use of perplexity as a utility metric, while relevant, may not capture the full spectrum of performance considerations. Perplexity is effective for general language model evaluation but may not align directly with task-specific outcomes, particularly for models fine-tuned for specialized applications. Evaluating additional utility metrics, such as task-specific accuracy or F1 scores for downstream tasks, would provide a more nuanced understanding of the impact of PEFT on utility.
To improve this, the author could expand the analysis to include a broader range of performance metrics, especially those relevant to specific tasks, offering a more comprehensive view of model utility and its trade-offs with privacy. This approach would be particularly beneficial for fine-tuning applications involving domain-specific tasks (e.g., question answering or sentiment analysis).

3. Language and Presentation Issues

Spelling Error: In Section 4.4, line 381, “measyured” should be corrected to “measured.”
Reference Issues: In Sections 4.2 and 4.3, there are references to Table 1, but the text mistakenly cites it as Table 3. These inconsistencies could lead to reader confusion.

4. Lack of Supporting Data

The analysis in Section 4.4, particularly the part discussing prompt-tuning, lacks direct data support or a clear reference to supplementary material in the appendix. This absence of detailed data makes it difficult for readers to validate the findings or understand the depth of the analysis. Suggested Improvement: Include specific data points within the main text or ensure they are readily accessible in the appendix for more transparent and verifiable results.

**Questions:**

Q1. In line 376, the presentation states: “The left figure shows results from models trained with differential privacy (DP), while the right figure presents results with DP at ϵ=10.0.” This is somewhat confusing as both figures are described as using DP. Could you clarify if the left figure represents models trained without DP? If both are indeed trained with DP, specifying the difference more explicitly would improve clarity.

Q2. The paper primarily uses perplexity as the utility metric, but perplexity may not reflect performance on specific downstream tasks. Would it be possible to include additional metrics, such as task-specific accuracy or F1 scores, to provide a broader view of PEFT’s utility? Alternatively, could the authors clarify if perplexity alone is considered a sufficient and representative utility measure across general scenarios?

---

> ### Author Response · Authors · 2024-11-24
> **Response to Reviewer CXB3**
>
> We first thank the reviewer for the time and effort in evaluating our work. We answer the reviewer’s concerns and questions below. We also updated our manuscript to reflect the reviewer’s valuable and constructive feedback.
>
> ----
>
> **[Weakness 1: Limited Scope of Privacy Attacks]**
>
> We thank the reviewer for their valuable comments on improving the accuracy of our claims within the scope of the privacy attacks. We are considering clarifying the focus of our work on data extraction attacks, which are arguably the most important concern in this context. Here we listed potential titles we can use, and we will also be happy to incorporate more suggestions from the reviewer:
>
> > Evaluating Memorization of Parameter-Efficient Fine-Tuning
>
> > Evaluating Membership Risks of Parameter-Efficient Fine-Tuning
>
>
> **[Weakness 2 and Question 2: Performance Metrics]**
>
> We first clarify that we adhere to standard practices for evaluating fine-tuned language models, as outlined in prior work [1]. In fine-tuning scenarios we employ, conventional evaluation metrics are not straightforward to apply. Metrics such as BLEU scores or accuracy on benchmarking datasets like Alpaca are designed to assess a model’s few-shot generation capabilities. On the other hand, task-specific accuracy can only be measured when we “fine-tune” pre-trained language models on specific benchmarking tasks (QA or sentiment analysis), not the tasks we use like MIMIC-III and/or Enron. However, we also acknowledge the importance of developing a universal metric for evaluating fine-tuned models across different datasets. We discuss this as a potential direction for future work in our conclusion.
>
> [1] Wen et al., Privacy Backdoors: Enhancing Membership Inference through Poisoning Pre-trained Models, NeurIPS 2024.
>
>
> **[Weakness 3, 4, and Question 1: Presentation Issues]**
>
> We thank the reviewer for pointing out mistakes we made in our manuscript. We first reviewed the sections presenting our main results to ensure there are no dangling references to the Appendix (e.g., we corrected the reference in Sec 4.4 to direct readers to the additional results on prompt-tuning, which are located at the end of the Appendix). Moreover, we proofread the manuscript to fix any typos (e.g., in Line 376, we intended to write "without DP," not "with DP").
>
> ----
>
> **[Summary]**
>
> We appreciate the reviewer’s constructive comments. If there is/are any remaining questions and concerns, we will be more than happy to address them during the discussion phase.

---

> > ### Comment · Reviewer_CXB3 · 2024-11-30
> > **confirmed**
> >
> > Thank you for your response. For Weakness 1, I believe the title "Evaluating Memorization of Parameter-Efficient Fine-Tuning" more accurately reflects your scope and contributions. For weakness 2, your response helped me understand clearly, which also reflected the importance of universal metric development. Thank you.

---

> ### Author Response · Authors · 2024-11-26
> **Kind Reminder of the Discussion Phase**
>
> Dear Reviewer CXB3;
>
> As the deadline for the discussion period approaches, we kindly ask if our response adequately addresses the reviewer’s concerns or if further clarification is needed. We look forward to hearing from the reviewer soon.
>
> Thank you!

---

### Official Review · Reviewer_yauG · 2024-11-03

**Soundness:** 4
**Presentation:** 4
**Contribution:** 4
**Rating:** 8
**Confidence:** 3

**Summary:**

The paper systematically studies how different parameter-efficient fine-tuning methods can memorize secrets. It uses a standard definition of "exposure" and evaluates how exposure is affected by PEFTs. Additionally, the paper studies how differential privacy can be applied to limit exposure for a limited utility loss.

The key contribution of this paper is its thorough evaluation, which uses two types of models (GPT2 and GPT2-XL) and 4 widespread PEFT techniques. The experiments carefully evaluate the impact of various PEFT configurations (algorithm, number of fine-tuning parameters, position of the secret) on memorization.

Some findings of the paper are surprising -- for instance, increasing the number of tunable parameters does not necessarily lead to an increase of memorization, in the context of PEFTs. The authors intend the paper to be a guide to privacy-preserving PEFTs ; I find that the paper satisfies this purpose.

**Strengths:**

Main strengths:
- The key contribution of this paper is its thorough evaluation of the PEFT landscape in terms of memorization. Experiments are extremely detailed, and provide extremely valuable datapoints for practitioners and future research.
- I really appreciated the focused and methodical approach of the paper. It made the space of PEFTs digestible (I am not an expert in PEFTs so I learned a lot). The evaluation questions are clearly-formulated and well-motivated.
- The paper does a great job breaking down a general question ("what are the privacy risks of PEFTs") into modular components that can be studied in a structured manner. Exposure, perplexity, and differential privacy budget are nicely orthogonal metrics. PEFT algorithm, number of tunable parameters, secret duplication and position are also great variables.

Other strengths:
- I appreciated the comparison of DP libraries in appendix. I didn't know about FastDP, so sharing this type of details is useful for other researchers.

**Weaknesses:**

Main weakness:
- I am concerned about the generalizability of the results to other types of secrets. The paper does a great job at varying the PEFT algorithms and parameters, but somehow the way secrets are chosen seems quite arbitrary.
- The authors hint at the fact that the type of secret can be an important variable ("We attribute this difference to the secrets we choose"). Thus, I find it surprising that this variable is not studied with the same depth as other variables. For instance, the author's hypothesis that the difference in exposure between Enron and MIMIC is due to secret types could be evaluated by also using an email address in MIMIC (and a name in Enron).
- I also wonder about other types of secrets, such as less common names, longer secrets, or numbers such as an SSN.


Other weaknesses:
- The fine-tuning datasets seem pretty small, with about 14k records (by the way, why is Enron 13,399 instead of exactly 13,431 records if the goal is to match the size of MIMIC?). I would be curious about the impact of dataset size on memorization, for instance on the Enron dataset with 600,000 records. But this might just have the inverse effect of secret duplication, so I understand if this variable is less important in the evaluation.
- Fig 4. "Setting ε below 10.0 completely breaks the models fine-tuned with prompt tuning and prefix tuning." Indeed, epsilon = 0.1 is probably too strong, so this might not be the most interesting value to pick. But what about epsilon=1? The scales of axes in Fig 4 are very different too.
- Fig 5: the privacy-utility tradeoff is pretty well understood, so I would personally be more interested in graphs showing epsilon vs memorization (and this paper is about memorization anyway). Fig 12 does that in appendix, but only for GPT2 XL, with just two datapoints. How about GPT2 for more values of epsilon?

Minor comments:
- I was looking for the equivalent of Fig 2 for prompt tuning, and finally found it in Fig 13. It could be helpful to include pointers to the appendix in the body of the paper.
- Table 1 is packed with useful information but a bit hard to digest, maybe put some numbers in bold?
- Why the quotation marks in: MEMORIZATION OF MODELS FINE-TUNED WITH “PRIVACY”
- typo" "Our definition above is strict: memorization is only confirmes"
- typo: "This construction fllows the same methodology as"
- typo: "insert the same secert at 5 different positions"

**Questions:**

* You formulate an interesting hypothesis about LoRA, saying that "the reduced rank in the latent representation space acts as an information bottleneck, making it difficult for the model to memorize outliers, such as the secret, which the model first encounters during fine-tuning (as we ensure the secret is not present in the pre-training corpus)." Have you evaluated this hypothesis by comparing how easily secrets can be memorized depending on how close they are compared to the pre-training dataset? For instance, what if a name is already present in the pre-training dataset, but the secret is about extracting the name in a particular context?
- What is the value of delta for the DP training? I assume you are not doing pure DP, but I couldn't find delta. Also, what is the unit of privacy for the DP training, is it token-level or record level?

---

> ### Author Response · Authors · 2024-11-24
> **Response to Reviewer yauG**
>
> We first thank the reviewer for the time and effort in evaluating our work. We answer the reviewer’s concerns and questions below. We also updated our manuscript to reflect the reviewer’s valuable and constructive feedback.
>
> ----
>
> **[Main Weakness: Generalizability]**
>
> We acknowledge the importance of evaluating the generalizability of our results across different secrets. In our preliminary investigation, we observed minimal variations as long as the secrets had not been encountered during pre-training. To provide more concrete evidence, we conducted the following additional experiments:
>
> **(Points 1 and 3: Less common names; other secret types)**
>
> We use the secret “clary zakharchuk” for fine-tuning GPT-2 models on the MIMIC-III data.
>
> |  |  | Base | Adapter | Prefix Tuning | Prompt Tuning | LoRA |
> |---|---|---|---|---|---|---|
> | MIMIC (GPT-2) | Exp. | 8.64 +/- 0.00 | 0.13 +/- 0.05 | 0.38 +/- 0.09 | 0.77 +/- 0.31 | 0.94 +/- 0.50 |
> |  | PPL. | 1.14 +/- 0.00 | 1.29 +/- 0.00 | 1.26 +/- 0.00 | 1.24 +/- 0.00 | 1.17 +/- 0.00 |
>
> The table above presents our results, which are consistent with the findings reported in our manuscript. Models fine-tuned using PEFT methods are less likely to memorize the secret.
>
>
> **(Points 1 and 2: Secrets from a completely different domains)**
>
> We also test with a secret that is unlikely to naturally occur in the fine-tuning dataset. We insert the secret “Leo.Moreno@gmail.com” into the MIMIC-III dataset composed of medical records.
>
> |  |  | Base | Adapter | Prefix Tuning | Prompt Tuning | LoRA |
> |---|---|---|---|---|---|---|
> | MIMIC (GPT-2) | Exp. | 8.64 +/- 0.00 | 2.92 +/- 1.70 | 1.20 +/- 0.59 | 0.46 +/- 0.15 | 0.68 +/- 0.35 |
> |  | PPL. | 1.14 +/- 0.00 | 1.29 +/- 0.00 | 1.26 +/- 0.00 | 1.24 +/- 0.00 | 1.17 +/- 0.00 |
>
> The table above summarizes our results, which are also consistent with the findings reported in our manuscript. Models fine-tuned using PEFT methods are less likely to memorize the secret. Prompt-tuning and LoRA achieve the lowest exposure, while the other two methods also reduce exposure to levels similar to the observation made in our manuscript.
>
> ----
>
> **[Other Weaknesses 1: Impact of the Dataset Size]**
>
> The reviewer is correct. We did not find any substantial impact of the dataset size on our findings in our initial investigation. We also agree with the reviewer that a quantitative illustration would present our results in a more convincing manner. Below we summarize the findings from our initial investigation, where we varied the size of the MIMIC-III dataset by increasing it by 100% (2x) and also decreasing it by randomly choosing 50% and 25% of the MIMIC-III dataset for our experiments.
>
> |  |  | Base | Adapter | Prefix Tuning | Prompt Tuning | LoRA |
> |---|---|---|---|---|---|---|
> | MIMIC (2x) | Exp. | 8.64 +/- 0.00 | 3.11 +/- 0.50 | 3.62 +/- 0.15 | 2.40 +/- 1.15 | 1.56 +/- 1.39 |
> |  | PPL. | 1.14 +/- 0.00 | 1.28 +/- 0.00 | 1.23 +/- 0.00 | 1.22 +/- 0.00 | 1.15 +/- 0.00 |
> | MIMIC (0.5x) | Exp. | 8.64 +/- 0.00 | 4.30 +/- 1.78 | 2.57 +/- 1.39 | 1.98 +/- 0.44 | 2.47 +/- 0.34 |
> |  | PPL. | 1.16 +/- 0.00 | 1.30 +/- 0.00 | 1.31 +/- 0.00 | 1.27 +/- 0.00 | 1.19 +/- 0.00 |
> | MIMIC (0.25x) | Exp. | 8.64 +/- 0.00 | 4.34 +/- 1.28 | 2.60 +/- 1.14 | 2.35 +/- 0.66 | 3.50 +/- 1.15 |
> |  | PPL. | 1.16 +/- 0.00 | 1.31 +/- 0.00 | 1.37 +/- 0.01 | 1.34 +/- 0.00 | 1.20 +/- 0.00 |
>
>
> Overall, the results remain consistent with those observed when we use the full dataset. Models fine-tuned with the PEFT mechanisms achieve lower memorization. Prompt-tuning and LoRA are the lowest, while Adapter and Prefix-tuning show slightly higher levels than the first two.
>
>
> **[Other Weaknesses 2 and 3: The Use of Other Epsilon Values]**
>
> - (W2) We acknowledge the reviewer’s concern about presenting only two epsilon values. Instead of presenting a figure like Figure 4, we decided to summarize our results in a table (see Table 2 in our revised manuscript) to clearly illustrate the interaction between exposure, perplexity, and various epsilon values across the different PEFT methods we examined.
>
> - (W3) We agree with the reviewer that evaluating GPT2-XL at a reasonable epsilon value (1.0), as suggested, could strengthen the validity of our results. Accordingly, we will conduct this additional experiment and make sure to add this to Figure 12 (Figure 11 in our revised manuscript) in our camera-ready version.
>
> ----
>
> continues to the next comment...

---

> ### Author Response · Authors · 2024-11-24
> **Response to Reviewer yauG (cont'd)**
>
> continues from the previous comment...
>
> ---
>
> **[Question 1: Information Bottleneck Hypothesis for LoRA]**
>
> We thank the reviewer for the suggestion. To evaluate the information bottleneck hypothesis for LoRA, we first ranked the perplexities of all candidate names used for MIMIC to identify the one that the model already exhibits a bias toward due to its pre-training procedure. We selected the name with the highest exposure without context in the pre-trained GPT-2 model. This name was then inserted into the fine-tuning dataset, and the model was fine-tuned with LoRA.
>
> Our findings show that the exposure is significantly higher when using this alternate name as the secret—up to 7.13, compared to 1.88 when "mary smith" is used as the secret. This supports the hypothesis that the biases of the pre-trained model and its dataset play a critical role in determining whether LoRA can memorize secrets in the fine-tuning dataset. These results are included in Appendix section B.12.
>
>
> **[Question 2: The Choice of Delta]**
>
> We use a record-level delta, calculated as the inverse of the dataset size. For both MIMIC and Enron, this delta is $\sim$7.4$\times$10$^{−7}$ (1/13.3k), following standard practices in prior work and the original study [1].
>
> [1] Abadi et al., Deep Learning with Differential Privacy, ACM CCS 2016
>
>
> ----
>
> **[Minor Comments]**
>
> We thank the reviewer for pointing out the presentation issues. We will also thoroughly review our manuscript again to ensure there are no remaining typos.
>
> ----
>
> **[Summary]**
>
> We appreciate the reviewer’s constructive comments. If there is/are any remaining questions and concerns, we will be more than happy to address them during the discussion phase.

---

> > ### Comment · Reviewer_yauG · 2024-11-26
> >
> > Thank you for the details, I appreciate the thorough experiments. And sounds good for delta.

---

### Official Review · Reviewer_wAdk · 2024-11-04

**Soundness:** 2
**Presentation:** 2
**Contribution:** 2
**Rating:** 5
**Confidence:** 2

**Summary:**

This paper examines privacy risks of models fine-tuned using various types of parameter-efficient fine-tuning. The risk assessment is performed by quantifying memorization of fine-tuning datasets through the Exposure metric. The authors conduct experiments on GPT-2 and GPT-2 XL using two datasets and four types of fine-tuning methods. Additionally, they investigate the impact of differential privacy on Exposure during parameter-efficient fine-tuning.

**Strengths:**

1. The paper addresses a timely research question about privacy vulnerabilities when using PEFT, which is particularly relevant given the widespread adoption of PEFT techniques in deploying LLMs across various domains.

2. The authors conducted extensive experiments and ablation studies to evaluate privacy leakage across various setups.

3. The findings demonstrating that PEFT methods offer significantly better privacy guarantees compared to full-parameter fine-tuning provide valuable insights for the broader ML community.

**Weaknesses:**

1. While this work is primarily experimental, the presentation quality of the experimental results requires improvement:

  * In Figures 1, 4, 5 (and similar figures in the Appendix), the axis scaling choices result in most data points being clustered in one corner, making it difficult to interpret the results accurately. This is particularly problematic in Figures 4 and 5, where high Perplexity values for a single point obscure the differences between other points. A recommended solution would be to revise the plotting approach (for instance, using log-scale (essentially, Cross-Entropy Loss) instead of raw Perplexity).

  * All figures in the main paper contain multiple plots, but their differences are only explained in the captions. This significantly impairs figure readability, as understanding the distinctions between plots requires repeatedly referring to the full description. One potential improvement would be to place key distinguishing features (such as $\varepsilon$ values in Figure 4) as titles above the corresponding plots.

  * The paper's analysis of DP privacy-utility trade-off would be more compelling if presented as plots with multiple points per method (such as Perplexity versus Exposure with points corresponding to different $\varepsilon$ values) rather than the current approach (Figure 4) of separate plots one by one with single point per method. (and with different scales and $\varepsilon$ values, which also makes direct comparisons challenging)

The main paper also contains strange results that were not commented on by the authors (see Questions).

2. In lines 246-247, the authors state that 'We observe a slight increase in perplexity (0.01-0.15), but the increase is too small to result in a significant increase in the exposure.' (Presumably, there's a typo and it should read 'decrease in exposure'.) This statement isn't entirely self-evident, given that the perplexity losses are quite substantial (~13%). To validate that such performance losses don't affect exposure, it would be valuable to measure exposure values for an undertrained baseline at perplexity levels matching those of the PEFT methods (e.g. for GPT-2 on MIMIC-III, at perplexity points of 1.30, 1.24, etc.).

3. In Definition 3.1, the authors introduce an adaptation of the term Memorization. The motivation for this decision is unclear since all experimental measurements use the Exposure metric, which, unlike Memorization, is adopted from [1] without modifications. The paper would benefit from a clearer explanation of why this adapted definition was included.

4. While the methods described in lines 126-131 don't provide formal guarantees, it would be beneficial to understand how their combination with PEFT affects privacy.

5. Additionally, the paper would benefit from explicit examples of dataset records after secret insertion (for example, in Appendix section).

[1] Carlini et. al., The secret sharer: Evaluating and testing unintended memorization in neural networks. In USENIX Security Symposium,
2019.

**Questions:**

1. How do the authors explain that in Figure 3, Exposure with differential privacy is worse than without it?

2. In Figure 3, why is the Exposure metric notably worse (higher) when using 1 insertion versus 500 insertions?

3. Given the high Exposure values with 500 insertions shown in Figure 2, was Table 1 constructed using 1 insertion? If so, this should be explicitly stated. Furthermore, it would be valuable to show the relationship between Exposure and intermediate insertion values (between 1 and 500).

4. In Figure 5, the dependencies on hyperparameters are not consistently monotonic (for example, $r=32$ is unexpectedly low in the top left, $r=16$ is low in the bottom left, and $pt=16$ between 32 and 64 in the bottom middle). Do the authors have any hypotheses explaining these inconsistencies?

---

> ### Author Response · Authors · 2024-11-24
> **Response to Reviewer wAdk**
>
> We thank the reviewer for the time and effort in reading and evaluating our manuscript. Below we answer the reviewer’s concerns and questions. We also updated our manuscript to reflect the constructive feedback from the reviewers.
>
> ----
>
> **[Weakness 1: Improving Presentations]**
>
> We thank the reviewer for the suggestions. We have made the following updates to our manuscript to improve clarity further:
>
> - Point 1: We revised Figure 1, 4, and 5 to use a log-scaled axis.
> - Point 2: We included titles to all the figures highlighting their distinguishing features.
> - Point 3: Instead of presenting our results in a figure (Figure 4, originally), we now use a table (Table 2) to clearly illustrate the interaction between exposure, perplexity, and epsilon across the PEFT methods we examined.
>
>
> **[Weakness 2: Additional Details about Our Results]**
>
> We first thank the reviewer for pointing out the typo in lines 246-247, where we indeed meant ‘decrease.’ We fixed it in our revised manuscript.
>
> We clarify that a reduction in exposure across our PEFT-trained models is *not* due to the inability to generalize well. To evaluate, we followed the reviewer's suggestion and measured the exposure of three undertrained models with perplexity similar to that observed in the PEFT-trained models. As shown below, all these undertrained models achieve an average exposure of 5.20–5.60, at least an order of magnitude greater than shown in Table 1 of our manuscript. We included the result in the Appendix and referenced it in Sec 4.4 to improve clarity.
>
> |                       | **Model 1**   | **Model 2**   | **Model 3**   |
> |-----------------------|---------------|---------------|---------------|
> | **Perplexity (PPL.)** | 1.17 +/- 0.00 | 1.25 +/- 0.00 | 1.35 +/- 0.00 |
> | **Exposure (Exp.)**   | 5.59 +/- 2.13 | 5.53 +/- 0.56 | 5.20 +/- 1.19 |
>
>
>
> **[Weakness 3: Motivation for Our Adaptation to the Memorization by Carlini et al.]**
>
> We adapted the memorization definition of Carlini et al. to examine the impact of secret positions within a training record. Carlini et al. focus exclusively on scenarios where a secret appears at the end of a training record, e.g., “Please contact me at **john.doe@gmail.com**” It does not consider many real-world cases where secrets appear in the middle of a context, such as “... since the prior **John Doe** radiograph, and a very small left apical…” Our evaluation reveals that depending on the design of the PEFT algorithms, a secret in certain positions is more likely to be memorized. This contrasts with the prior work, which shows that secrets at the end of a sentence are more likely to be memorized.
>
>
> **[Weakness 4: Combination of Empirical Privacy Defenses and PEFT]**
>
> We believe that this topic is beyond the scope of our work and could constitute a separate study. Hence, this wouldn’t be our weakness. For instance, prior work [1] has examined the impact of a subset of such heuristic protection methods, such as data augmentation, on membership inference attacks against computer vision models. However, we acknowledge the importance of future research in empirically evaluating heuristic protection mechanisms for privacy across different PEFT methods and have included this as a potential future work in our conclusion.
>
> [1] Kaya et al., When Does Data Augmentation Help With Membership Inference Attacks, ICLR 2021
>
>
> **[Weakness 5: Illustrating Example Records in Evaluation]**
>
> We have included examples of secrets inserted into training records in the Appendix of our revised manuscript.
>
> ----
>
> continued in the next comment

---

> ### Author Response · Authors · 2024-11-24
> **Response to Reviewer wAdk (cont'd)**
>
> continued from the previous comment.
>
> ----
>
> **[Questions 1 and 2: Clarification of Our Results in Figure 3]**
>
> We admit our mistake in including an incorrect figure: the left figure represents an average exposure computed across different seeds, while the right figure is from a single observation. We fixed the right figure. The revised figure now shows that exposure is lower when trained with DP and that 500 insertions lead to higher exposure values compared to a single insertion.
>
>
> **[Question 3: Number of Insertions in (1, 500)]**
>
> We excluded results for insertion counts between 1 and 500 because we did not observe any meaningful variations in exposure. For instance, with 5 insertions, both standard fine-tuning and PEFT methods typically reached the maximum exposure in most cases. The only case where we observed a variation was when the secret was located at the first token position. To help understand, we provide an example of this variation using standard fine-tuning on GPT-w with the MIMIC dataset:
>
>
> | **# Insertion**     | **1** | **10** | **50** | **100** | **500** |
> |---------------------|-------|--------|--------|---------|---------|
> | **Exposure (Exp.)** | 6.05  | 7.37   | 8.09   | 8.25    | 8.51    |
>
>
> **[Question 4: Impact of PEFT Hyper-parameters Under DP]**
>
> We could not find any consistent relationship between PEFT hyper-parameters and the perplexity under DP. The first observation we had is that the trend differs from the datasets we use. In the MIMIC-III dataset, larger hyperparameters generally result in higher perplexity (except for Adapter with $r=32$). In contrast, we observe a reduction in perplexity for the Enron dataset under the same conditions. One possible explanation could be that there are optimal PEFT hyper-parameters required to achieve reasonable performance (e.g., in MIMIC-III, the rank $\sim$4--8 and the prefix tokens $\sim$16). Increasing those parameters beyond the optimal range can increase the noise added by DP-SGD and make the performance fluctuate.
>
> ----
>
> **[Summary]**
>
> We appreciate the reviewer’s feedback on the clarity and presentation issues.
>
> The current rating for our work is 3, which, according to the review guidelines of other premier ML conference venues, typically indicates "a paper with technical flaws, weak evaluation, inadequate reproducibility, and/or incompletely addressed ethical considerations." We see the weaknesses in the current review do not fall into these categories; thus, we kindly request the reviewer to reconsider the rating if our answers clarify existing issues. If there is/are any remaining concern(s), we will be happy to address them during the discussion phase.

---

> ### Author Response · Authors · 2024-11-26
> **Kind Reminder of the Discussion Phase**
>
> Dear Reviewer wAdk;
>
> As the deadline for the discussion period approaches, we kindly ask if our response adequately addresses the reviewer’s concerns or if further clarification is needed. We look forward to hearing from the reviewer soon.
>
> Thank you!

---

> ### Comment · Reviewer_wAdk · 2024-11-29
> **Follow-up questions**
>
> I thank the authors for the detailed response. Most of my concerns have been addressed, and I have increased the score to 5. However, I still have several remaining concerns:
>
> **Regarding Weakness 3:**
>
> My concern was that introducing a Definition for a term that is barely used later in the paper (e.g. not used in the experiments) somewhat overloads the paper. In my opinion, describing the differences between the experimental setup and Carlini et al. could have been sufficiently covered in just one or several paragraphs.
>
> **Regarding Figure 3:**
>
> If the experiments in Figure 3 and Table 2 were conducted under the same setup (I couldn't find specific model and dataset details for Table 2, though I might have missed them), then Table 2 suggests that at $\varepsilon=10.0$, the model's perplexity was already above 10, indicating severely degraded model quality. I request that the authors clarify the setup for Table 2 and specify the model's perplexity in Figure 3 at $\varepsilon=10.0$, as I believe examining other metrics (such as exposure) is not meaningful when the model's performance is unsatisfactory.
>
> **Regarding Question 3:**
>
> I believe such an ablation study could be beneficial to demonstrate how exposure deteriorates with increasing insertions. If exposure reaches its maximum value at 5 insertions (as stated in the authors' response), it would be valuable to examine the behavior for insertions between 1 and 5. Furthermore, it seems illogical to conduct most experiments with 500 insertions when exposure reaches similar values at just 5 insertions.

---

> ### Author Response · Authors · 2024-12-03
> **Response to Follow-up Questions**
>
> Dear Reviewer wAdk,
>
> We sincerely thank the reviewers for the continuous engagement in the discussion.
>
> **[Memorization Definition (Weakness 3)]**
>
> We acknowledge the importance of clarifying the connection between our memorization definition and our experimental designs. We will make sure to spare one (or a few) paragraphs in the experimental setup section of our camera-ready version to address this.
>
> ----
>
> **[Prefix-tuning Results Shown in the Left-figure of Figure 3]**
>
> We clarify the experimental setup produced the results in Table 2 and Figure 3. In Table 2 (for prefix-tuning), we set the number of prefix tokens to 64. But for the analysis shown in Figure 3, we consistently set all the PEFT hyper-parameters to 16, including the number of prefix tokens. The dataset we use in Table 2 is the same as for our main results in Table 1. We stated this setup in lines 424–425; but, we acknowledge that improving the clarity of this description will enhance the readability of our results. We will make this clear in our camera-ready version.
>
> We acknowledge that prefix-tuning is *not* particularly compatible with differential privacy. In Figure 3, while the model perplexity does not reach as high as 10, it remains relatively high at 5.0–5.2. To address this concern, we ran prefix-tuning with an epsilon of 160, which ensures the model perplexity of 1.5, comparable to other PEFT methods at an epsilon of 10. Even in this setup, we find a trend consistent with our results in Figure 3: the first few tokens are more likely to be memorized when prefix-tuning is applied. Near position 0, we observe exposure of ~2.5, while near position 50, exposures are ~1.2. We will make sure to present our results when the prefix-tuned models achieve a reasonable perplexity in the final version.
>
> ----
>
> **[Impact of the Number of Secret Insertions (Question 3)]**
>
> We clarify that this evaluation aims to determine “whether the trend observed with a single insertion holds when the number of insertions increases to 500—an extreme case tested in prior work [1], where memorization of a secret can almost always be ensured.” One insertion represents the trend likely to be observed in practice (as the number of duplications is unlikely to be 500). But with 500 insertions, following the upper bound of the worst-case duplication of a secret as studied in prior work [1], we ensure that the observation from a single insertion is not coincidental but is a consistent pattern. Tightening the number of insertions based on empirical observations risks the evaluation of our hypotheses above.
>
> We acknowledge that the evaluation suggested by the reviewer could address a different research question: “How do the memorization behaviors of PEFT-ed models vary with different numbers of insertions?” Our answer from the evaluation with the number of insertions {1, 10, 50, 100, 500} is “We do not observe any significant variations.” But examining with more fine-grained intervals such as in (1, 4) to answer this question is beyond the scope of this work.
>
> ----
>
> We hope our response has addressed the reviewer’s concerns. We know that the deadline for the discussion phase is approaching, but If there is/are any remaining questions and concerns, we will be more than happy to address them.
>
> Sincerely,
>
> The Authors of the Submission 11024
>
> ----
>
> [1] Carlini et al. Quantifying Memorization Across Neural Language Models, ICLR 2023

---

> ### Comment · Reviewer_wAdk · 2024-12-03
>
> I thank the authors for the detailed response. I acknowledge that I have read it.
>
> A few concluding comments:
>
> * I don't see a description of the dataset and model in lines 424-425; there is only a discussion on the PEFT hyperparameters and the $\varepsilon$ value.
>
> * I also wasn't aware that the setup with 500 follows prior work; it would be beneficial to specify this in the paper.
>
> * Additionally, I find some of the authors' arguments regarding my previous concerns not convincing enough.
>
> Therefore, I am maintaining the score of 5.

---

### Author Response · Authors · 2024-12-04
**Summary Response**

We sincerely thank the reviewers for their valuable feedback. The discussion phase has been very productive, and we were able to greatly enhance the clarity of our evaluation results and deliver our findings in a more convincing manner through additional empirical observations.

Based on the comments from each reviewer, we believe that there are no longer any significant concerns about our work. We would like to highlight our novel findings, which we believe are of the community’s potential interest in the field of machine learning and privacy studies.



**[Summary of Our Contributions and Impacts]**



**Significance**

Our work, through an extensive evaluation, studies the privacy implications of a new emerging paradigm of efficiently fine-tuning ever-increasing-scale language models. This can be potentially important as these PEFT mechanisms are increasingly prevalent across various domains and actively being deployed to user-facing applications. The practice of fine-tuning large public models on small, often confidential or proprietary datasets presents a scenario in which privacy concerns often become paramount. As a result of their much smaller size, PEFT modules can be saved, shared, and stored publicly with much more ease than fully-finetuned models, increasing the accessibility of these models to potential adversaries. Due to their very recent and rapid development and adoption, there exists little prior work into the unique privacy risks of PEFT. We begin to fill that gap with this work.

**Technical Contributions**

We systematically approached this problem and made a series of findings specific to the models fine-tuned using PEFT methods. PEFT-trained models have significantly less vulnerability to data extraction attacks compared to the full standard fine-tuning. Prefix-tuning memorizes secrets in a pattern opposite to prior work focusing on memorization when using full-model fine-tuning. This finding demonstrates that depending on the fine-tuning method one uses, there are specific risks to secrets in the first few tokens rather than the last few (as established in prior work). As a result, we warn practitioners against assuming universal secret “safety” based on position. When it comes to training under a privacy guarantee, out of all the fine-tuning methods, we find consistently the best privacy-utility trade-off comes from LoRA. This may suggest efficient fine-tuning methods based around reparameterization (rather than adding new modules – something the other PEFT methods do) most effectively leverage knowledge and capabilities in the pretrained model while at the same time avoiding significant memorization of fine-tune data.

**Potential community implications**

We provide implications to three entities concerning the PEFT mechanisms: researchers, practitioners and algorithm designers.
- For researchers, the implications are that empirical privacy leakage can be minimized with moderate privacy guarantees and thus guarantees below epsilon=10 generally degrade the privacy-utility trade-off. In addition, the reparameterization-based method we investigated, LoRA, demonstrated superior privacy-utility trade-off to the other three additive methods. Our result suggests that reparamterization-based PEFT should be the focus of researchers interested in an optimal privacy-utility trade-off.
- For practitioners interested in fine-tuning and deploying models to production, we show that efficient fine-tuning can serve as a viable replacement for other privacy-preserving methods, while providing comparable performance. If deployment requires fine-tuning with a privacy guarantee, across all PEFT, LoRA will provide the best utility under a range of privacy budgets, while maintaining low privacy leakage. Nevertheless, choosing the right architecture and hyperparameters is a non-trivial task, especially in settings where a privacy guarantee is required (and where performance can be sensitive to PEFT hyperparameters). Practitioners should therefore be prepared to devote resources finding optimal hyperparameter/architecture combinations for such cases.
- For developers we observe that even small changes within a particular PEFT architecture can have significant and unexpected effects in both model memorization and performance, and should be tuned carefully with these metrics in mind. In addition, it may be necessary to directly optimize architecture and hyperparameters for DP-compatibility depending on the fine-tune data domain. Thus, developers should potentially be prepared to tailor architectures and their hyperparameters to specific dataset and privacy scenarios, rather than relying solely on designs which perform well on benchmark datasets without private training.


We hope this summary is helpful in the post-rebuttal discussion.

Sincerely,

The Authors of the Submission 11024

---

### Meta-Review · Area_Chair_Ys3w · 2024-12-23

**Metareview:**

This paper studies the impact of parameter-efficient fine-tuning (PEFT) for language models on privacy, focusing on susceptibility to data extraction attacks. The authors evaluate four popular PEFT methods: adapters, prefix-tuning, prompt-tuning, and LoRA. They also consider two datasets: the MIMIC-III (health records) and the Enron email dataset. These methods are assessed through experiments on GPT-2 and GPT-2 XL, using exposure as the primary metric for memorization and perplexity as a proxy for utility. The authors also study the use of DP-SGD to control privacy risks. The findings suggest that PEFT-trained models generally demonstrate lower privacy risks than standard fine-tuning.

The reviewers recognized the importance of the topic and appreciated some of the findings in the experiments. However, they also highlighted several critical issues regarding the clarity and presentation of the experimental results, which dilutes the paper's contribution.

For instance, Reviewer Vj3Z highlighted that the paper primarily focuses on a "secret guess" task and questioned if the findings generalize to larger-scale models. Reviewer wAdk noted several points of confusion related to definitions and experiments, ranging from the way figures are formatted to fundamental issues surrounding the methodology (e.g., repeating an experiment 5 times != than training on a new language model each time), raising concerns about the generalizability of the findings. This concern was echoed by reviewer CXB3, which noted the limited scope of privacy attacks (e.g., missing membership inference attacks), though they remained more positive post-rebuttal.

Given that this paper is largely experimental and critical weaknesses were identified surrounding the experimental methodology -- ranging from scope, grounding of claims, and clarity of results from the experiments -- I believe the paper is not yet ready for publication. This is a clear "major revision" case, with the paper needing an additional round of review. The authors are encouraged to address the lack of clarity in their experimental design, provide more comprehensive evaluations, and refine the presentation for future submissions. The reviewers' comments will significantly strengthen the paper.

**Additional Comments On Reviewer Discussion:**

There was an extensive back-and-forth between reviewers and authors. Post-rebuttal, several questions remained surrounding the experimental methodology used in the paper.

---

### Decision · Program_Chairs · 2025-01-22

Reject